# Strategic Preys Make Acute Predators: Enhancing Camouflaged Object Detectors by Generating Camouflaged Objects

**Chunming He**[1] , **Kai Li**[2*], **Yachao Zhang**[1] , **Yulun Zhang**[3] ,
**Chenyu You**[4] , **Zhenhua Guo**[5] , **Xiu Li**[1*], **Martin Danelljan**[6] , **Fisher Yu**[6]
[1]Shenzhen International Graduate School, Tsinghua University,
[2]NEC Laboratories America, [3]Shanghai Jiao Tong University, [4]Yale University,
[5]Tianyi Traffic Technology, [6]ETH Zürich,

## Abstract

Camouflaged object detection (COD) is the challenging task of identifying camouflaged objects visually blended into surroundings. Albeit achieving remarkable success, existing COD detectors still struggle to obtain precise results in some challenging cases. To handle this problem, we draw inspiration from the prey-vs-predator game that leads preys to develop better camouflage and predators to acquire more acute vision systems and develop algorithms from both the prey side and the predator side. On the prey side, we propose an adversarial training framework, Camouflageator, which introduces an auxiliary generator to generate more camouflaged objects that are harder for a COD method to detect. Camouflageator trains the generator and detector in an adversarial way such that the enhanced auxiliary generator helps produce a stronger detector. On the predator side, we introduce a novel COD method, called Internal Coherence and Edge Guidance (ICEG), which introduces a camouflaged feature coherence module to excavate the internal coherence of camouflaged objects, striving to obtain more complete segmentation results. Additionally, ICEG proposes a novel edge-guided separated calibration module to remove false predictions to avoid obtaining ambiguous boundaries. Extensive experiments show that ICEG outperforms existing COD detectors and Camouflageator is flexible to improve various COD detectors, including ICEG, which brings state-of-the-art COD performance. The code will be available at https://github.com/ChunmingHe/Camouflageator.

## 1 Introduction

The never-ending prey-vs-predator game drives preys to develop various escaping strategies. One of the most effective and ubiquitous strategies is camouflage. Preys use camouflage to blend into the surrounding environment, striving to escape hunting from predators. For survival, predators, on the other hand, must develop acute vision systems to decipher camouflage tricks. Camouflaged object detection (COD) is the task that aims to mimic predators' vision systems and localize foreground objects that have subtle differences from the background. The intrinsic similarity between camouflaged objects and the backgrounds renders COD a more challenging task than traditional object detection (Liu et al., 2020), and has attracted increasing research attention for its potential applications in medical image analysis (Tang et al., 2023) and species discovery (He et al., 2023b).

Traditional COD solutions (Hou & Li, 2011; Pan et al., 2011) mainly rely on manually designed strategies with fixed extractors and thus are constrained by limited discriminability. Benefiting from the powerful feature extraction capacity of convolutional neural network, a series of deep learning-based methods have been proposed and have achieved remarkable success on the COD task (He et al., 2023c;d; Zhai et al., 2022). However, when facing some extreme camouflage scenarios, those methods still struggle to excavate sufficient discriminative cues crucial to *precisely* localize objects of interest. For example, as shown in the top row of Fig. 1, the state-of-the-art COD method, FEDER

---

*Corresponding Author, † The work was mainly done when Yulun Zhang was at ETH Zürich.

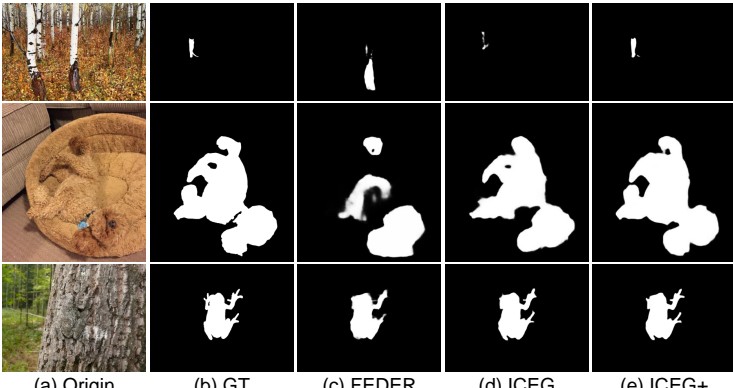

(a) Origin     (b) GT     (c) FEDER     (d) ICEG     (e) ICEG+

Figure 1: Results of FEDER (He et al., 2023c), ICEG, and ICEG+. ICEG+ indicates to optimize ICEG under the Camouflageator framework. Both ICEG and ICEG+ generate more complete results with clearer edges. ICEG+ also exhibits better localization capacity.

(He et al., 2023c), cannot even roughly localize the object and thus produce a completely wrong result. Sometimes, even though a rough position can be obtained, FEDER still fails to precisely segment the objects, as shown in the two remaining rows of Fig. 1. While FEDER manages to find the rough regions for the objects, the results are either incomplete (middle row: some key parts of the dog are missing) or ambiguous (bottom row: the boundaries of the frog are not segmented out).

This paper aims to address these limitations. We are inspired by the prey-vs-predator game, where preys develop more deceptive camouflage skills to escape predators, which, in turn, pushes the predators to develop more acute vision systems to discern the camouflage tricks. This game leads to ever-strategic preys and ever-acute predators. With this inspiration, we propose to address COD by developing algorithms on both the prey side that generates more deceptive camouflage objects and the predator side that produces complete and precise detection results.

On the prey side, we propose a novel adversarial training framework, Camouflageator, which generates more camouflaged objects that make it even harder for existing detectors to detect and thus enhance the generalizability of the detectors. Specifically, as shown in Fig. 2, Camouflageator comprises an auxiliary generator and a detector, which could be any existing detector. We adopt an alternative two-phase training mechanism to train the generator and the detector. In Phase I, we freeze the detector and train the generator to synthesize camouflaged objects aiming to deceive the detector. In Phase II, we freeze the generator and train the detector to accurately segment the synthesized camouflaged objects. By iteratively alternating Phases I and II, the generator and detector both evolve, helping to obtain better COD results.

On the predator side, we present a novel COD detector, termed Internal Coherence and Edge Guidance (ICEG), which particularly aims to address the issues of incomplete segmentation and ambiguous boundaries of existing COD detectors. For incomplete segmentation, we introduce a camouflaged feature coherence (CFC) module to excavate the internal coherence of camouflaged objects. We first explore the feature correlations using two feature aggregation components, *i.e.*, the intralayer feature aggregation and the contextual feature aggregation. Then, we propose a camouflaged consistency loss to constrain the internal consistency of camouflaged objects. To eliminate ambiguous boundaries, we propose an edge-guided separated calibration (ESC) module. ESC separates foreground and background features using attentive masks to decrease uncertainty boundaries and remove false predictions. Besides, ESC leverages edge features to adaptively guide segmentation and reinforce the feature-level edge information to achieve the sharp edge for segmentation results. We integrate the Camouflageator framework with ICEG to get ICEG+, which can exhibit better localization capacity (see Fig. 1). Our contributions are summarized as follows:

- We design an adversarial training framework, Camouflageator, for the COD task. Camouflageator employs an auxiliary generator that generates more camouflaged objects that are harder for COD detectors to detect and hence enhances the generalizability of those detectors. Camouflageator is flexible and can be integrated with various existing COD detectors.

- We propose a new COD detector, ICEG, to address the issues of incomplete segmentation and ambiguous boundaries that existing detectors face. ICEG introduces a novel CFC mod-

ule to excavate the internal coherence of camouflaged objects to obtain complete segmentation results, and an ESC module to leverage edge information to get precise boundaries.

- Experiments on four datasets verify that Camouflageator can promote the performance of various existing COD detectors, ICEG significantly outperforms existing COD detectors, and integrating Camouflageator with ICEG reaches even better results.

## 2 RELATED WORK

### 2.1 CAMOUFLAGED OBJECT DETECTION

Traditional methods rely on hand-crafted operators with limited feature discriminability (He et al., 2019), failing to handle complex scenarios. A Bayesian-based method (Zhang et al., 2016) was proposed to separate the foreground and background regions through camouflage modeling. Learning-based approaches have become mainstream in COD with three main categories: *(i) Multi-stage framework:* SegMaR (Jia et al., 2022) was the first plug-and-play method to integrate segment, magnify, and reiterate under a multi-stage framework. However, SegMaR has limitations in flexibility due to not being end-to-end trainable. *(ii) Multi-scale feature aggregation:* PreyNet (Zhang et al., 2022) proposed a bidirectional bridging interaction module to aggregate cross-layer features with attentive guidance. UGTR (Yang et al., 2021) proposed a probabilistic representational model combined with transformers to explicitly address uncertainties. DTAF (Ren et al., 2021) developed multiple texture-aware refinement modules to learn the texture-aware features. Similarly, FGANet (Zhai et al., 2022) designed a collaborative local information interaction module to aggregate structure context features. *(iii) Joint training strategy:* MGL (Zhai et al., 2021) designed the mutual graph reasoning to model the correlations between the segmentation map and the edge map. BGNet (Sun et al., 2022) presented a joint framework for COD to detect the camouflaged candidate and its edge using a cooperative strategy. Analogously, FEDER (He et al., 2023c) jointly trained the edge reconstruction task with the COD task and guided the segmentation with the predicted edge.

We improve existing methods in three aspects: *(i)* Camouflageator is the first end-to-end trainable plug-and-play framework for COD, thus ensuring flexibility. *(ii)* ICEG is the first COD detector to alleviate incomplete segmentation by excavating the internal coherence of camouflaged objects. *(iii)* Unlike existing edge-based detectors (Sun et al., 2022; He et al., 2023c; Xiao et al., 2023), ICEG employs edge information to guide segmentation adaptively under the separated attentive framework.

### 2.2 ADVERSARIAL TRAINING

Adversarial training is a widely-used solution with many applications, including adversarial attack (Zhang et al., 2021) and generative adversarial network (GAN) (Deng et al., 2022; Li et al., 2020). Recently, several GAN-based methods have been proposed for the COD task. JCOD (Li et al., 2021) introduced a GAN-based framework to measure the prediction uncertainty. ADENet (Xiang et al., 2021) employed GAN to weigh the contribution of depth for COD. Distinct from those GAN-based methods, our Camouflageator enhances the generalizability of existing COD detectors by generating more camouflaged objects that are harder to detect.

## 3 METHODOLOGY

When preys develop more deceptive camouflaged skills to escape predators, the predators respond by evolving more acute vision systems to discern the camouflage tricks. Drawing inspiration from this prey-vs-predator game, we propose to address COD by developing the Camouflageator and ICEG techniques that mimic preys and predators, respectively, to generate more camouflaged objects and to more accurately detect the camouflaged objects, improving the generalizability of the detector.

### 3.1 CAMOUFLAGEATOR

Camouflageator is an adversarial training framework that employs an auxiliary generator $G_c$ to synthesize more camouflaged objects that make it even harder for existing detectors $D_s$ to detect and thus enhance the generalizability of the detectors. We train $G_c$ and $D_s$ alternatively in a two-phase adversarial training scheme. Fig. 2 shows the framework.

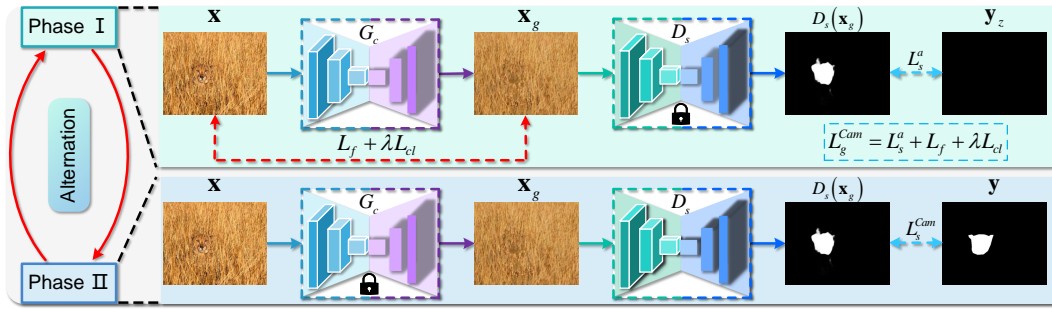

Figure 2: Architecture of Camouflageator. In Phase I, we fix detector $D_s$ and update generator $G_c$ to synthesize more camouflaged objects to deceive $D_s$. In Phase II, we fix $G_c$ and train the detector $D_s$ to segment the synthesized image.

**Training the generator.** We fix the detector $D_s$ and train the generator $G_c$ to generate more deceptive objects that fail the detector. Given a camouflaged image $\mathbf{x}$, we generate

$$\mathbf{x}_g = G_c(\mathbf{x}), \tag{1}$$

and expect that $\mathbf{x}_g$ is more deceptive to $D_s$ than $\mathbf{x}$. To achieve this, $\mathbf{x}_g$ should be visually consistent (similar in global appearance) with $\mathbf{x}$ but simultaneously have those discriminative features crucial for detection hidden or reduced.

To encourage visual consistency, we propose to optimize the fidelity loss represented as follows:

$$L_f = \| (\mathbf{1}-\mathbf{y})\otimes\mathbf{x}_g - (\mathbf{1}-\mathbf{y})\otimes\mathbf{x}\|^2, \tag{2}$$

where $\mathbf{y}$ is the ground truth binary mask and $\otimes$ denotes element-wise multiplication. Since $(\mathbf{1}-\mathbf{y})$ denotes the background mask, this term in essence encourage $\mathbf{x}_g$ to be similar with $\mathbf{x}$ for the background region. We encourage fidelity by preserving only the background rather than the whole image because otherwise, it hinders the generation of camouflaged objects in the foreground.

To hide discriminative features, we optimize the following concealment loss to imitate the bio-camouflage strategies, *i.e.*, internal similarity and edge disruption (Price et al., 2019), as

$$L_{cl} = \|\mathbf{y}\otimes\mathbf{x}_g - P_o^I\|^2 + \|\mathbf{y}_e\otimes\mathbf{x}_g - P_e^I\|^2, \tag{3}$$

where $\mathbf{y}_e$ is the weighted edge mask dilated by Gaussian function (Jia et al., 2022) to capture richer edge information. $P_o^I$ is the image-level object prototype which is an average of foreground pixels. $P_e^I$ is the image-level edge prototype which is an average of edge pixels specified by $\mathbf{y}_e$. Note that $\mathbf{y}_e$, $P_o^I$, and $P_e^I$ are all derived from the provided ground truth $\mathbf{y}$ and help to train the model. This term encourages individual pixels of the foreground region and the edge region of $\mathbf{x}_g$ to be similar to the average values, which has a smooth effect and thus hides discriminative features.

Apart from the above concealment loss, we further employ the detector $D_s$ to reinforce the concealment effect. The idea is that if $\mathbf{x}_g$ is perfectly deceptive, $D_s$ tends to detect nothing as the foreground. To this end, we optimize

$$L_s^a = L_{BCE}^w \left(D_s\left(\mathbf{x}_g\right), \mathbf{y}_z\right) + L_{IoU}^w \left(D_s\left(\mathbf{x}_g\right), \mathbf{y}_z\right), \tag{4}$$

where $\mathbf{y}_z = \mathbf{0}$ is an all-zero mask. $L_{BCE}^w(\cdot)$ and $L_{IoU}^w(\cdot)$ denote the weighted binary cross-entropy loss (Jadon, 2020) and the weighted intersection-over-union loss (Rahman & Wang, 2016).

By introducing a trade-off parameter $\lambda$, our overall learning objective to train $G_c$ is as follows,

$$L_g^{Cam} = L_s^a + L_f + \lambda L_{cl}. \tag{5}$$

**Training the detector.** In Phase **II**, we fix the generator $G_c$ and train the detector $D_s$ to accurately segment the synthesized camouflaged objects. This is the standard COD task and various existing COD detectors can be employed, for example, the simple one we used above,

$$L_s^{Cam} = L_{BCE}^w \left(D_s\left(\mathbf{x}_g\right), \mathbf{y}\right) + L_{IoU}^w \left(D_s\left(\mathbf{x}_g\right), \mathbf{y}\right). \tag{6}$$

## 3.2 ICEG

We further propose ICEG to alleviate incomplete segmentation and eliminate ambiguous boundaries. Given $\mathbf{x}$ of size $W \times H$, we start by using a basic encoder $F$ to extract a set of deep features $\{f_k\}_{k=0}^4$ with the resolution of $\frac{W}{2^{k+1}} \times \frac{H}{2^{k+1}}$ and employ ResNet50 (He et al., 2016) as the default architecture.

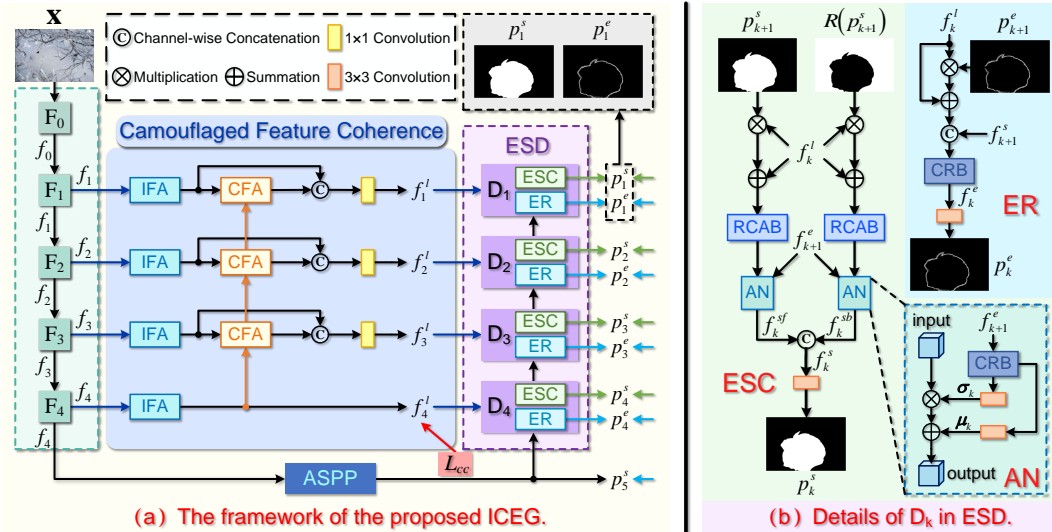

Figure 3: Framework of our ICEG. CRB is the Conv-ReLU-BN structure. We omit the Sigmoid operator in (b) for clarity.

As shown in Fig. 3, we then feed these features, i.e., $\{f_k\}_{k=1}^4$, to the camouflaged feature coherence (CFC) module and the edge-guided segmentation decoder (ESD) for further processing. Moreover, the last feature map $f_4$, which has rich semantic cues, is fed into an atrous spatial pyramid pooling (ASPP) module $A_s$ (Yang et al., 2018) and a $3 \times 3$ convolution $conv3$ to generate a coarse result $p_5^s$: $p_5^s = conv3(A_s(f_4))$, where $p_5^s$ shares the same spatial resolution with $f_4$.

### 3.2.1 CAMOUFLAGED FEATURE COHERENCE MODULE

To alleviate incomplete segmentation, we propose the camouflaged feature coherence (CFC) module to excavate the internal coherence of camouflaged objects. CFC consists of two feature aggregation components, i.e., the intra-layer feature aggregation (IFA) and the contextual feature aggregation (CFA), to explore feature correlations. Besides, CFC introduces a camouflaged consistency loss to constrain the internal consistency of camouflaged objects.

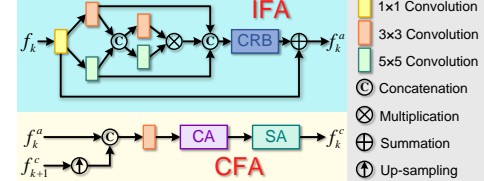

Figure 4: Details of IFA and CFA.

**Intra-layer feature aggregation.** In Fig. 4, IFA seeks the feature correlations by integrating the multi-scale features with different reception fields in a single layer, assuring that the aggregated features can capture scale-invariant information. Given $f_k$, a $1 \times 1$ convolution $conv1$ is first applied for channel reduction, followed by two parallel convolutions with different kernel sizes. This process produces the features $f_k^3$ and $f_k^5$ with varying receptive fields:

$$f_k^3 = conv3(conv1(f_k)), f_k^5 = conv5(conv1(f_k)), \quad (7)$$

where $conv5$ is $5 \times 5$ convolution. Then we combine $f_k^3$ and $f_k^5$, process them with two parallel convolutions, and multiply the outputs to excavate the scale-invariant information:

$$f_k^{35} = conv3\big(conca\big(f_k^3, f_k^5\big)\big) \otimes conv5\big(conca\big(f_k^3, f_k^5\big)\big), \quad (8)$$

where $conca(\cdot)$ denote concatenation. We then integrate the three features and process them with a CRB block $CRB(\cdot)$, i.e., $3 \times 3$ convolution, ReLU, and batch normalization. By summing with the channel-wise down-sampled feature, the aggregated features $\{f_k^a\}_{k=1}^4$ are formulated as follows:

$$f_k^a = conv1(f_k) + CRB\big(conca\big(f_k^3, f_k^5, f_k^{35}\big)\big). \quad (9)$$

**Contextual feature aggregation.** CFA explores the inter-layer feature correlations by selectively interacting cross-level information with channel attention and spatial attention (Woo et al., 2018), which ensures the retention of significant coherence. The aggregated feature $\{f_k^c\}_{k=1}^3$ is

$$f_k^c = SA\big(CA\big(conv3\big(conca\big(up\big(f_{k+1}^c\big), f_k^a\big)\big)\big)\big), \quad (10)$$

where $up(\cdot)$ is up-sampling operation. $CA(\cdot)$ and $SA(\cdot)$ are channel attention and spatial attention. $f_4^c = f_4^a$. Given $\{f_k^c\}_{k=1}^3$, the integrated features $\{f_k^l\}_{k=1}^3$ conveyed to the decoder are

$$f_k^l = conv1\big(concate\big(f_k^a, f_k^c\big)\big). \quad (11)$$

We employ $conv1$ for channel integration and $f_4^l = f_4^a$.

**Camouflaged consistency loss.** To enforce the internal consistency of the camouflaged object, we propose a camouflaged consistency loss to enable more compact internal features. To achieve this, one intuitive idea is to decrease the variance of the camouflaged internal features. However, such a constraint can lead to feature collapse, *i.e.*, all extracted features are too clustered to be separated, thus diminishing the segmentation capacity. Therefore, apart from the above constraint, we propose an extra requirement to keep the internal and external features as far away as possible. We apply the feature-level consistency loss to the deepest feature $f_4^l$ for its abundant semantic information:

$$L_{cc} = \|\mathbf{y}_d \otimes f_4^l - P_o^f\|^2 - \|\mathbf{y}_d \otimes f_4^l - P_b^f\|^2, \tag{12}$$

where $\mathbf{y}_d$ is the down-sampled ground truth mask. $P_o^f$ and $P_b^f$ denote the feature-level prototypes of the camouflaged object and the background, respectively.

**Discussions.** Apart from focusing on feature correlations as in existing detectors (Zhang et al., 2022; He et al., 2023c), we design a novel camouflaged consistency loss to enhance the internal consistency of camouflaged objects, facilitating complete segmentation.

### 3.2.2 EDGE-GUIDED SEGMENTATION DECODER

As depicted in Fig. 3, edge-guided segmentation decoder (ESD) $\{D_k\}_{k=1}^4$ comprises an edge reconstruction (ER) module and an edge-guided separated calibration (ESC) module to generate the edge predictions $\{p_k^e\}_{k=1}^4$ and the segmentation results $\{p_k^s\}_{k=1}^5$, respectively.

**Edge reconstruction module.** We introduce an ER module to reconstruct the object boundary. Assisted by the edge map $p_{k+1}^e$ and the segmentation feature $f_{k+1}^s$ from the former decoder, the edge feature $f_k^e$ is presented as follows:

$$f_k^e = CRB(conca(f_k^l \otimes p_{k+1}^e + f_k^l, f_{k+1}^s)). \tag{13}$$

where $f_5^s = A_s(f_4)$ and $p_k^e = conv3(f_k^e)$. $f_5^e$ and $p_5^e$ are set as zero for initialization. We repeat $p_{k+1}^e$ as a 64-dimension tensor to ensure channel consistency with $f_k^l$ in Eq. (13).

**Edge-guided separated calibration module.** Ambiguous boundary, a common problem in COD, manifests as two phenomena: (1) a high degree of uncertainty in the fringes, and (2) the unclear edge of the segmented object. We have observed that the high degree of uncertainty is mainly due to the intrinsic similarity between the camouflaged object and the background. To address this issue, we separate the features from the foreground and the background by introducing the corresponding attentive masks, and design a two-branch network to process the attentive features. This approach helps decrease uncertainty fringes and remove false predictions, including false-positive and false-negative errors. Given the prediction map $p_{k+1}^s$, the network is defined as follows:

$$f_k^s = conca(f_k^{sf}, f_k^{sb}), p_k^s = conv3(f_k^s), \tag{14}$$

where $f_k^{sf}$ and $f_k^{sb}$ are the foreground and the background attentive features, which are formulated

$$f_k^{sf} = RCAB\left(f_k^l \otimes S\left(p_{k+1}^s\right) + f_k^l\right), \tag{15a}$$

$$f_k^{sb} = RCAB\left(f_k^l \otimes S\left(R\left(p_{k+1}^s\right)\right) + f_k^l\right), \tag{15b}$$

where $S(\cdot)$ and $R(\cdot)$ are Sigmoid and reverse operators, *i.e.*, element-wise subtraction with 1. $RCAB(\cdot)$ is the residual channel attention block (Zhang et al., 2018), which is used to emphasize those informative channels and high-frequency information.

The second phenomenon, unclear edge, is due to the extracted features giving insufficient importance to edge information. In this case, we explicitly incorporate edge features to guide the segmentation process and promote edge prominence. Instead of simply superimposing, we design an adaptive normalization (AN) strategy with edge features to guide the segmentation in a variational manner, which reinforces the feature-level edge information and thus ensures the sharp edge of the segmented object. Given the edge feature $f_{k+1}^e$, the attentive features can be acquired by:

$$f_k^{sf} = \boldsymbol{\sigma}_k^f \otimes (RCAB(f_k^l \otimes S(p_{k+1}^s) + f_k^l)) + \boldsymbol{\mu}_k^f, \tag{16a}$$

$$f_k^{sb} = \boldsymbol{\sigma}_k^b \otimes (RCAB(f_k^l \otimes S(R(p_{k+1}^s)) + f_k^l)) + \boldsymbol{\mu}_k^b, \tag{16b}$$

where $\{\boldsymbol{\sigma}_k^f, \boldsymbol{\mu}_k^f\}$ and $\{\boldsymbol{\sigma}_k^b, \boldsymbol{\mu}_k^b\}$ are variational parameters. In AN, $\{\boldsymbol{\sigma}_k, \boldsymbol{\mu}_k\}$ can be calculated by:

$$\boldsymbol{\sigma}_k = conv3_\sigma(CRB_\sigma(f_{k+1}^e)), \boldsymbol{\mu}_k = conv3_\mu(CRB_\mu(f_{k+1}^e)). \tag{17}$$

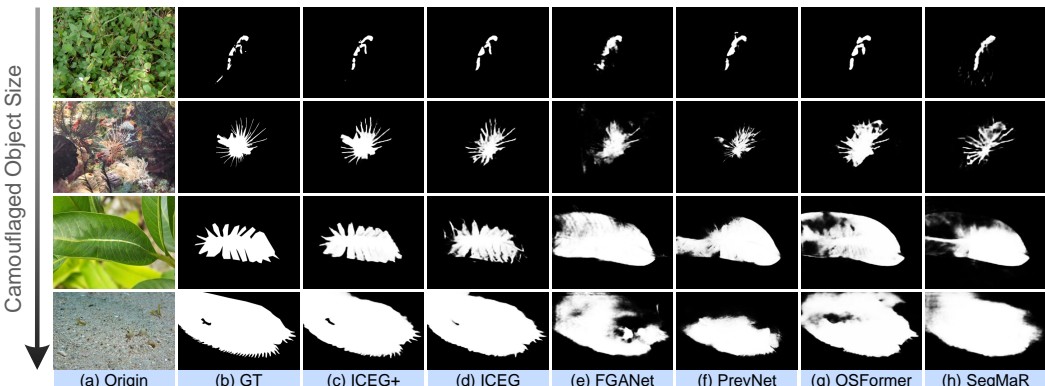

Figure 5: Qualitative analysis of ICEG and other four cutting-edge methods. ICEG generates more complete results with clearer edges. We also provide the results of ICEG+, which is optimized under Camouflageator.

**Discussions.** Unlike existing edge-guided methods (Sun et al., 2022; He et al., 2023c) that focus only on edge guidance, we combine edge guidance with foreground/background splitting using attentive masks. This integration enables us to decrease uncertainty fringes and remove false predictions along edges, thus achieving the sharp edge for segmentation results.

### 3.2.3 LOSS FUNCTIONS OF ICEG

Apart from the camouflaged consistency loss $L_{cc}$, our ICEG is also constraint with the segmentation loss $L_s$ and the edge loss $L_e$ to supervise the segmentation results $\{p_k^s\}_{k=1}^5$ and the reconstructed edge results $\{p_k^e\}_{k=1}^4$. Following (Fan et al., 2021), we define $L_s$ as

$$L_s = \sum_{k=1}^{5} \frac{1}{2^{k-1}} \left( L_{BCE}^w \left( p_k^s, \mathbf{y} \right) + L_{IoU}^w \left( p_k^s, \mathbf{y} \right) \right). \tag{18}$$

For edge supervision, we employ dice loss $L_{dice}(\cdot)$ (Milletari et al., 2016) to overcome the extreme imbalance in edge maps:

$$L_e = \sum_{k=1}^{4} \frac{1}{2^{k-1}} L_{dice} \left( p_k^e, \mathbf{y}_e \right). \tag{19}$$

Therefore, with the assistance of a trade-off parameter $\beta$, the total loss is presented as follows:

$$L_t = L_s + L_e + \beta L_{cc}. \tag{20}$$

### 3.2.4 ICEG+

To promote the adoption of our Camouflageator, we provide a use case and utilize ICEG+ to denote the algorithm that integrates our Camouflageator framework with ICEG. The integration is straightforward; we only need to replace the detector supervision from Eq. (6) with Eq. (20). In addition, we pre-train ICEG with $L_t$ (Eq. (20)) to ensure the training stability. See Sec. 4.1 for more details.

## 4 EXPERIMENTS

### 4.1 EXPERIMENTAL SETUP

**Implementation details.** All experiments are implemented on PyTroch on two RTX3090 GPUs. For Camouflageator, the generator adopts ResUNet as its backbone. As for ICEG, a pre-trained ResNet50 (He et al., 2016) on ImageNet (Krizhevsky et al., 2017) is employed as the default encoder. We also report the COD results with other encoders, including Res2Net50 (Gao et al., 2019) and Swin Transformer (Liu et al., 2021). Following (Fan et al., 2020), we resize the input image as $352 \times 352$ and pre-train ICEG by Adam with momentum terms $(0.9, 0.999)$ for 100 epochs. The batch size is set as 36 and the learning rate is initialized as 0.0001, decreased by 0.1 every 50 epochs. Then we use the same batch size to further optimize ICEG under the Camouflageator framework for 30 epochs and get ICEG+, where the optimizer is Adam with parameters $(0.5, 0.99)$ and the initial learning rate is 0.0001, dividing by 10 every 15 epochs. $\lambda$ and $\beta$ are set as 0.1.

| Methods | Backbones | CHAMELEON | | | | CAMO | | | | COD10K | | | | NC4K | | | |
|---|---|---|---|---|---|---|---|---|---|---|---|---|---|---|---|---|---|
| | | $M\downarrow$ | $F_\beta\uparrow$ | $E_\phi\uparrow$ | $S_\alpha\uparrow$ | $M\downarrow$ | $F_\beta\uparrow$ | $E_\phi\uparrow$ | $S_\alpha\uparrow$ | $M\downarrow$ | $F_\beta\uparrow$ | $E_\phi\uparrow$ | $S_\alpha\uparrow$ | $M\downarrow$ | $F_\beta\uparrow$ | $E_\phi\uparrow$ | $S_\alpha\uparrow$ |
| Common Setting: Single Input Scale and Single Stage | | | | | | | | | | | | | | | | | |
| SegMaR-1 (Jia et al., 2022) | ResNet50 | 0.028 | 0.828 | 0.944 | 0.892 | 0.072 | 0.772 | 0.861 | 0.805 | 0.035 | 0.699 | 0.890 | 0.813 | 0.052 | 0.767 | 0.885 | 0.835 |
| PreyNet (Zhang et al., 2022) | ResNet50 | 0.027 | 0.844 | 0.948 | 0.895 | 0.077 | 0.763 | 0.854 | 0.790 | 0.034 | 0.715 | 0.894 | 0.813 | 0.047 | 0.798 | 0.887 | 0.838 |
| FGANet (Zhai et al., 2022) | ResNet50 | 0.030 | 0.838 | 0.944 | 0.896 | 0.070 | 0.769 | 0.865 | 0.800 | 0.032 | 0.708 | 0.894 | 0.803 | 0.047 | 0.800 | 0.891 | 0.837 |
| FEDER (He et al., 2023c) | ResNet50 | 0.028 | 0.850 | 0.944 | 0.892 | 0.070 | 0.775 | 0.870 | 0.802 | 0.032 | 0.715 | 0.892 | 0.810 | 0.046 | 0.808 | 0.900 | 0.842 |
| ICEG (Ours) | ResNet50 | 0.027 | 0.858 | 0.950 | 0.899 | 0.068 | 0.789 | 0.879 | 0.810 | 0.030 | 0.747 | 0.906 | 0.826 | 0.044 | 0.814 | 0.908 | 0.849 |
| PreyNet+ (Ours) | ResNet50 | 0.027 | 0.856 | 0.954 | 0.901 | 0.074 | 0.778 | 0.869 | 0.808 | 0.031 | 0.744 | 0.908 | 0.833 | 0.044 | 0.821 | 0.912 | 0.859 |
| FGANet+ (Ours) | ResNet50 | 0.029 | 0.847 | 0.948 | 0.898 | 0.069 | 0.781 | 0.877 | 0.814 | 0.030 | 0.735 | 0.911 | 0.823 | 0.045 | 0.814 | 0.905 | 0.854 |
| FEDER+ (Ours) | ResNet50 | 0.027 | 0.855 | 0.947 | 0.895 | 0.068 | 0.793 | 0.883 | 0.820 | 0.030 | 0.739 | 0.905 | 0.831 | 0.043 | 0.820 | 0.910 | 0.845 |
| ICEG+ (Ours) | ResNet50 | 0.026 | 0.863 | 0.952 | 0.903 | 0.066 | 0.805 | 0.891 | 0.829 | 0.028 | 0.763 | 0.920 | 0.843 | 0.041 | 0.835 | 0.922 | 0.869 |
| SINet V2 (Fan et al., 2021) | Res2Net50 | 0.030 | 0.816 | 0.942 | 0.888 | 0.070 | 0.779 | 0.882 | 0.822 | 0.037 | 0.682 | 0.887 | 0.815 | 0.048 | 0.792 | 0.903 | 0.847 |
| BGNet (Sun et al., 2022) | Res2Net50 | 0.029 | 0.835 | 0.944 | 0.895 | 0.073 | 0.744 | 0.870 | 0.812 | 0.033 | 0.714 | 0.901 | 0.831 | 0.044 | 0.786 | 0.907 | 0.851 |
| ICEG (Ours) | Res2Net50 | 0.025 | 0.869 | 0.958 | 0.908 | 0.066 | 0.808 | 0.903 | 0.838 | 0.028 | 0.752 | 0.914 | 0.845 | 0.042 | 0.828 | 0.917 | 0.867 |
| ICEG+ (Ours) | Res2Net50 | 0.023 | 0.873 | 0.960 | 0.910 | 0.064 | 0.826 | 0.912 | 0.845 | 0.026 | 0.770 | 0.925 | 0.853 | 0.040 | 0.844 | 0.928 | 0.878 |
| ICON (Zhuge et al., 2022) | Swin | 0.029 | 0.848 | 0.940 | 0.898 | 0.058 | 0.794 | 0.907 | 0.840 | 0.033 | 0.720 | 0.888 | 0.818 | 0.041 | 0.817 | 0.916 | 0.858 |
| ICEG (Ours) | Swin | 0.023 | 0.860 | 0.959 | 0.905 | 0.044 | 0.855 | 0.926 | 0.867 | 0.024 | 0.782 | 0.930 | 0.857 | 0.034 | 0.855 | 0.932 | 0.879 |
| ICEG+ (Ours) | Swin | 0.022 | 0.867 | 0.961 | 0.908 | 0.042 | 0.861 | 0.931 | 0.871 | 0.023 | 0.788 | 0.934 | 0.862 | 0.033 | 0.861 | 0.937 | 0.883 |
| Other Setting: Multiple Input Scales (MIS) | | | | | | | | | | | | | | | | | |
| ZoomNet (Pang et al., 2022) | ResNet50 | 0.024 | 0.858 | 0.943 | 0.902 | 0.066 | 0.792 | 0.877 | 0.820 | 0.029 | 0.740 | 0.888 | 0.838 | 0.043 | 0.814 | 0.896 | 0.853 |
| ICEG (Ours) | ResNet50 | 0.023 | 0.864 | 0.957 | 0.905 | 0.063 | 0.802 | 0.889 | 0.833 | 0.028 | 0.751 | 0.913 | 0.840 | 0.042 | 0.827 | 0.911 | 0.873 |
| Other Setting: Multiple Stages (MS) | | | | | | | | | | | | | | | | | |
| SegMaR-4 (Jia et al., 2022) | ResNet50 | 0.025 | 0.855 | 0.955 | 0.906 | 0.071 | 0.779 | 0.865 | 0.815 | 0.033 | 0.737 | 0.896 | 0.833 | 0.047 | 0.793 | 0.892 | 0.845 |
| ICEG-4 (Ours) | ResNet50 | 0.024 | 0.870 | 0.961 | 0.907 | 0.067 | 0.802 | 0.884 | 0.823 | 0.028 | 0.755 | 0.920 | 0.843 | 0.043 | 0.824 | 0.915 | 0.860 |

Table 1: Quantitative comparisons of ICEG and other 13 SOTAs on four benchmarks. SegMaR-1 and SegMaR-4 are SegMaR at one stage and four stages. "+" indicates optimizing the detector under our Camouflageator framework. Swin and PVT denote Swin Transformer and PVT V2. The best results are marked in **bold**. For ResNet50 backbone in the common setting, the best two results are in red and blue fonts.

| Metrics | Effect of $L_f$ and $L_{cl}$ | | | Effect of AT strategy | | |
|---|---|---|---|---|---|---|
| | w/o $L_g^c$ | w/ $L_f$ | w/ $L_g^c$ | BO w/o $L_{cl}$ | BO w/ $L_g^c$ | AT w/ $L_g^c$ |
| $M\downarrow$ | 0.032 | 0.030 | 0.028 | 0.030 | 0.032 | 0.028 |
| $F_\beta\uparrow$ | 0.721 | 0.750 | 0.763 | 0.752 | 0.722 | 0.763 |
| $E_\phi\uparrow$ | 0.899 | 0.907 | 0.920 | 0.910 | 0.895 | 0.920 |
| $S_\alpha\uparrow$ | 0.816 | 0.834 | 0.843 | 0.832 | 0.812 | 0.843 |

(a) Origin  (b) w/o $L_g^c$  (c) w/ $L_f$  (d) w/ $L_g^c$

Figure 6: Synthesized images of the generator trained by different losses.

Table 2: Ablation study of Camouflageator on *COD10K*. $L_g^c = L_f + \lambda L_{cl}$. BO and AT are bi-level optimization and adversarial training.

**Datasets.** We use four COD datasets for evaluation, including *CHAMELEON* (Skurowski et al., 2018), *CAMO* (Le et al., 2019), *COD10K* (Fan et al., 2021), and *NC4K* (Lv et al., 2021). *CHAMELEON* comprises 76 camouflaged images. *CAMO* contains 1,250 images with 8 categories. *COD10K* has 5,066 images with 10 super-classes. *NC4K* is the largest test set with 4,121 images. Following the common setting (Fan et al., 2020; 2021), our training set involves 1,000 images from *CAMO* and 3,040 images from *COD10K*, and our test set integrates the rest from the four datasets.

**Metrics.** Following previous methods (Fan et al., 2020; 2021), we employ four commonly-used metrics, including mean absolute error $(M)$, adaptive F-measure $(F_\beta)$, mean E-measure $(E_\phi)$, and structure measure $(S_\alpha)$. Note that smaller $M$ or larger $F_\beta$, $E_\phi$, $S_\alpha$ signify better performance.

## 4.2 COMPARISON WITH THE STATE-OF-THE-ARTS

**Quantitative analysis.** We compare our ICEG with 13 state-of-the-art (SOTA) solutions in three different settings. Apart from the common setting, two other settings (multiple input scales and multiple stages) are also included, where ICEG follows the corresponding practices of ZoomNet (Pang et al., 2022) and SegMaR (Jia et al., 2022). As shown in Table 1, ICEG outperforms the SOTAs by a large margin in all settings and backbones. In the common setting, ICEG overall surpasses the second-best methods in 2.1%, 5.2%, 8.3% with the backbone of ResNet50 (FEDER (He et al., 2023c)), Res2Net50 (BGNet (Sun et al., 2022)), Swin Transformer (ICON (Zhuge et al., 2022)). Moreover, we also present the results of detectors optimized under Camouflageator. In Table 1, Camouflageator generally improves other detectors by 2.8% (PreyNet), 2.2% (FGANet), 2.3% (FEDER), and increases our ICEG by 2.5% (ResNet50), 2.3% (Res2Net50), 1.4% (Swin Transformer), which verifies that our Camouflageator is a plug-and-play framework. Results of the compared methods are generated by their provided models with the image size of $352 \times 352$ for fairness.

**Qualitative analysis.** Fig. 5 shows that ICEG gets more complete results than existing methods, especially for large objects whose intrinsic correlations are more dispersed (the last row). This substantiates the effectiveness of the our CFC module that excavates the internal coherence of cam-

| Metrics | Ablation study of CFC component | | | | | | Ablation study of ESD component | | | | | | Ours |
|---|---|---|---|---|---|---|---|---|---|---|---|---|---|
| | w/o CFC | w/o IFA | w/o CFA | w/o FA | w/o $L_{cc}$ | $L_{cc}$->$L_{cc}^1$ | w/o ESD | w/o ESC | SC->FC | SC->BC | w/o AN | w/o ER | ICEG |
| $M \downarrow$ | 0.035 | 0.032 | 0.031 | 0.033 | 0.032 | 0.032 | 0.035 | 0.034 | 0.032 | 0.031 | 0.033 | 0.034 | **0.030** |
| $F_\beta \uparrow$ | 0.685 | 0.728 | 0.731 | 0.720 | 0.722 | 0.704 | 0.678 | 0.688 | 0.737 | 0.741 | 0.715 | 0.693 | **0.747** |
| $E_\phi \uparrow$ | 0.866 | 0.885 | 0.893 | 0.883 | 0.887 | 0.890 | 0.864 | 0.871 | 0.896 | 0.902 | 0.890 | 0.872 | **0.906** |
| $S_\alpha \uparrow$ | 0.808 | 0.814 | 0.822 | 0.812 | 0.816 | 0.813 | 0.802 | 0.806 | 0.820 | 0.822 | 0.815 | 0.804 | **0.826** |

Table 3: Ablation study of ICEG on *COD10K*. "–>" is substitution. (a) FA includes both IFA and CFA. $L_{cc}^1$ is the first term of $L_{cc}$ in Eq. (12). (b) SC, FC, BC are short for separated (Eq. (16)), foreground (Eq. (16a)), background (Eq. (16b)) calibration. Note that "w/o ER" removes edge predictions, thus including "w/o AN".

ouflaged objects for generating more complete prediction maps. Moreover, ICEG gets clearer edges for the predictions than the existing methods, thanks to our ESD module that decreases uncertainty fringes and eliminates unclear edges of the segmented object. Moreover, we can see that ICEG+ obtains even better results than ICEG, further verifying the effect of our Camouflageator framework.

## 4.3 ABLATION STUDY AND ANALYSIS

**Validity of Camouflageator.** We conduct validity analyses for Camouflageator, including our objective function in Eq. (5) and the adversarial training manner. As shown in Fig. 6, the generator trained without $L_g^c$ produces the images with severe artifacts, while the one trained with fidelity loss $L_f$ only synthesizes visual-appealing images but fails to hide discriminative features. In contrast, the generator trained with our $L_g^c$ generates high-quality images with more camouflaged objects, ensuring the generalizability of the detector (see Table 2 ). We also compare Camouflageator with the bi-level optimization (BO) framework (He et al., 2023a) to verify the advancement of our adversarial manner. BO involves the auxiliary generator and the detector in both the training and testing phases without adversarial losses, *i.e.*, Eqs. (4) and (6). As the concealment loss $L_{cl}$ may limit the performance in such an end-to-end manner, we also report the results optimized without $L_{cl}$, namely with only $L_f$ and segmentation loss (Eq. (20)). Table 2 verifies the effect of our adversarial manner.

**Effect of CFC and ESD.** The efficacy of CFC modules is verified in Table 3. In Table 3, we examine the impact of the CFC module (in (a)) and investigate the effect of individual components in CFC, including feature aggregation components (in (b), (c), and (d)), and $L_{cc}$ (in (e) and (f)). As shown in Table 3 (e), our camouflaged consistency loss $L_{cc}$ generally improves our detector by $3.4\%$, which its positive effect. Furthermore, we demonstrate the superiority of $L_{cc}$ by incorporating $L_{cc}$ into existing cutting-edge detectors, as detailed in the supplementary materials. Additionally, we present detailed ablation results for ESD in Table 3, where we highlight the benefits of ESD, ESC, separated calibration, adaptive normalization, and the joint strategy to integrate the ER task into the COD task. Moreover, as observed in Table 3, the combination of edge guidance with foreground/background splitting using attentive masks is shown to further boost detection performance. Such discovery can bring insights for the design of edge guidance modules.

## 5 CONCLUSION

In this paper, we propose to address COD on both the prey and predator sides. On the prey side, we introduce a novel adversarial training strategy, Camouflageator, to enhance the generalizability of the detector by generating more camouflaged objects harder for a COD detector to detect. On the predator side, we design a novel detector, dubbed ICEG, to address the issues of incomplete segmentation and ambiguous boundaries. In specific, ICEG employs the CFC module to excavate the internal coherence of camouflaged objects and applies the ESD module for edge prominence, thus producing complete and precise detection results. Extensive experiments verify our superiority.

**Acknowledgements.** This work is supported by National Key R&D Program of China (Grant No. 2020AAA0108303), Shenzhen Science and Technology Project (Grant No. JCYJ20200109143041798). Shenzhen Stable Supporting Program (WDZC20200820200655001). Shenzhen Key Laboratory of next generation interactive media innovative technology (Grant No. ZDSYS 20210623092001004). The authors express their appreciation to Dr. Fengyang Xiao for her insightful comments, improving the quality of this paper.

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

# Supplementary Materials for Strategic Preys Make Acute Predators: Enhancing Camouflaged Object Detectors by Generating Camouflaged Objects

**Chunming He**[1] , **Kai Li**[2*], **Yachao Zhang**[1] , **Yulun Zhang**[3] ,
**Chenyu You**[4] , **Zhenhua Guo**[5], **Xiu Li**[1*], **Martin Danelljan**[6], **Fisher Yu**[6]
[1]Shenzhen International Graduate School, Tsinghua University,
[2]NEC Laboratories America, [3]Shanghai Jiao Tong University, [4]Yale University,
[5]Tianyi Traffic Technology, [6]ETH Zürich,

## Contents

| Methods | Backbones | CHAMELEON (76 images) | | | | CAMO (250 images) | | | | COD10K (2,026 images) | | | | NC4K (4,121 images) | | | |
|---|---|---|---|---|---|---|---|---|---|---|---|---|---|---|---|---|---|
| | | $M\downarrow$ | $F_\beta\uparrow$ | $E_\phi\uparrow$ | $S_\alpha\uparrow$ | $M\downarrow$ | $F_\beta\uparrow$ | $E_\phi\uparrow$ | $S_\alpha\uparrow$ | $M\downarrow$ | $F_\beta\uparrow$ | $E_\phi\uparrow$ | $S_\alpha\uparrow$ | $M\downarrow$ | $F_\beta\uparrow$ | $E_\phi\uparrow$ | $S_\alpha\uparrow$ |
| Common Setting: Single Input Scale and Single Stage | | | | | | | | | | | | | | | | | |
| HitNet (Hu et al., 2023) | PVT | 0.024 | 0.861 | 0.944 | 0.907 | 0.060 | 0.791 | 0.892 | 0.834 | 0.027 | 0.790 | 0.922 | 0.847 | 0.042 | 0.825 | 0.911 | 0.858 |
| ICEG (Ours) | PVT | **0.022** | **0.879** | **0.957** | **0.913** | **0.043** | **0.863** | **0.933** | **0.876** | **0.022** | **0.805** | **0.938** | **0.871** | **0.030** | **0.869** | **0.941** | **0.890** |
| ICEG+ (Ours) | PVT | 0.022 | 0.881 | 0.958 | 0.915 | 0.041 | 0.867 | 0.935 | 0.877 | 0.021 | 0.811 | 0.939 | 0.873 | 0.029 | 0.875 | 0.944 | 0.893 |

Table S1: Quantitative comparisons of ICEG and HitNet (Hu et al., 2023) on four benchmarks. "+" indicates optimizing the detector under our Camouflageator framework. PVT denotes PVT V2 (Wang et al., 2022). The best results are marked in **bold**.

## A  Comparison with the state-of-the-arts

**Quantitative analysis.** We provide the results of ICEG using the PVT V2 backbone (Wang et al., 2022) and compare it with HitNet (AAAI2023) (Hu et al., 2023). As illustrated in Table S1, ICEG outperforms HitNet by $9.6\%$, demonstrating the superiority of our ICEG. Furthermore, we also present the results of ICEG+ and discover that the Camouflageator framework generally improves ICEG performance by $1.0\%$ on average in PVT V2, thus further validating the efficacy of our Camouflageator framework.

## B  Ablation study and analysis

We conduct the ablation study on the *COD10k* dataset.

**Ablation study of fidelity loss and concealment loss.** In the manuscript, we apply the fidelity loss $L_f$ only to the background rather than the whole image. This is because imposing the fidelity loss on the entire image may hinder the generation of more concealed objects in the foreground. In order to verify the effect of our fidelity loss, we conduct an experiment, and the results are shown in Table S2. In this experiment, the fidelity loss in $L_g^{c1}$ is imposed on the whole image and the rest components in $L_g^{c1}$ are the same as those in $L_g^c$, where $L_g^c = L_f + \lambda L_{cl}$. $L_g^{c1}$ is defined as:

$$L_g^{c1} = \|\mathbf{x}_g - \mathbf{x}\|^2 + \lambda L_{cl}. \tag{1}$$

---

*Corresponding Author, † The work was mainly done when Yulun Zhang was at ETH Zürich.

| Methods | $M \downarrow$ | $F_\beta \uparrow$ | $E_\phi \uparrow$ | $S_\alpha \uparrow$ |
|---|---|---|---|---|
| w/ $L_g^c$ | **0.028** | **0.763** | **0.920** | **0.843** |
| $L_g^c \rightarrow L_g^{c1}$ | 0.029 | 0.754 | 0.917 | 0.838 |
| $L_g^c \rightarrow L_g^{c2}$ | 0.029 | 0.758 | 0.915 | 0.836 |
| w/o Cam | 0.030 | 0.747 | 0.906 | 0.826 |

Table S2: Effect of our fidelity loss and concealment loss, where $L_g^c = L_f + \lambda L_{cl}$, "–>" means substitution, "w/" and "w/o" denote with and without. Cam is short for Camouflageator.

| Methods | $M \downarrow$ | $F_\beta \uparrow$ | $E_\phi \uparrow$ | $S_\alpha \uparrow$ |
|---|---|---|---|---|
| $352 \times 352$ | 0.030 | 0.747 | 0.906 | 0.826 |
| $384 \times 384$ | 0.030 | 0.749 | 0.911 | 0.826 |
| $512 \times 512$ | 0.028 | 0.755 | 0.915 | 0.843 |
| $702 \times 702$ | **0.027** | **0.762** | **0.919** | **0.848** |

Table S3: Ablation study of ICEG at different spatial resolutions.

| Methods | $M \downarrow$ | $F_\beta \uparrow$ | $E_\phi \uparrow$ | $S_\alpha \uparrow$ |
|---|---|---|---|---|
| w/ ASPP | **0.030** | **0.747** | **0.906** | **0.826** |
| w/o ASPP | **0.030** | 0.745 | 0.903 | 0.824 |

Table S4: Effect of ASPP.

| Methods | $M \downarrow$ | $F_\beta \uparrow$ | $E_\phi \uparrow$ | $S_\alpha \uparrow$ |
|---|---|---|---|---|
| SegMaR-1 (Jia et al., 2022) | 0.035 | 0.699 | 0.890 | 0.813 |
| SegMaR-1+$L_{cc}$ | 0.034 | 0.716 | 0.895 | 0.816 |
| PreyNet (Zhang et al., 2022a) | 0.034 | 0.715 | 0.894 | 0.813 |
| PreyNet+$L_{cc}$ | 0.032 | 0.734 | 0.898 | 0.819 |
| FGANet (Zhai et al., 2022) | 0.032 | 0.708 | 0.894 | 0.803 |
| FGANet+$L_{cc}$ | 0.032 | 0.721 | 0.896 | 0.808 |
| FEDER (He et al., 2023b) | 0.032 | 0.715 | 0.892 | 0.810 |
| FEDER+$L_{cc}$ | 0.031 | 0.728 | 0.899 | 0.817 |
| ICEG-R50 (Ours) | 0.030 | 0.747 | 0.906 | 0.826 |

Table S5: Effect of our camouflaged consistency loss $L_{cc}$ with the backbone of ResNet50, where R50 denotes ResNet50 (He et al., 2016).

| Methods | $M \downarrow$ | $F_\beta \uparrow$ | $E_\phi \uparrow$ | $S_\alpha \uparrow$ |
|---|---|---|---|---|
| MGL (Zhai et al., 2021) | 0.035 | 0.680 | 0.851 | 0.814 |
| MGL+ESD | 0.034 | 0.687 | 0.854 | 0.816 |
| FEDER (He et al., 2023b) | 0.032 | 0.715 | 0.892 | 0.810 |
| FEDER+ESD | 0.031 | 0.722 | 0.896 | 0.814 |
| ICEG-R50 (Ours) | 0.030 | 0.747 | 0.906 | 0.826 |
| BG-Net (Sun et al., 2022) | 0.033 | 0.714 | 0.901 | 0.831 |
| BG-Net+ESD | 0.032 | 0.725 | 0.904 | 0.835 |
| ICEG-R2N (Ours) | 0.028 | 0.752 | 0.914 | 0.845 |

Table S6: Comparison with other edge-based methods with the same backbone, where R50 and R2N denote ResNet50 (He et al., 2016) and Res2Net50 (Gao et al., 2019), respectively.

| Methods | $M \downarrow$ | $F_\beta \uparrow$ | $E_\phi \uparrow$ | $S_\alpha \uparrow$ |
|---|---|---|---|---|
| (0,0,0,0) | 0.034 | 0.688 | 0.871 | 0.806 |
| (1,0,0,0) | 0.032 | 0.710 | 0.884 | 0.814 |
| (1,1,0,0) | 0.031 | 0.723 | 0.892 | 0.817 |
| (1,1,1,0) | **0.030** | 0.734 | 0.902 | 0.822 |
| (1,1,1,1) | **0.030** | **0.747** | **0.906** | **0.826** |

Table S7: Effect of the ESC module.

| Methods | $M \downarrow$ | $F_\beta \uparrow$ | $E_\phi \uparrow$ | $S_\alpha \uparrow$ |
|---|---|---|---|---|
| R50 w/o Cam | 0.030 | 0.747 | 0.906 | 0.826 |
| R50 w/o pretrain | 0.029 | 0.755 | 0.913 | 0.835 |
| R50 w/ pretrain | **0.028** | **0.763** | **0.920** | **0.843** |

| Methods | $M \downarrow$ | $F_\beta \uparrow$ | $E_\phi \uparrow$ | $S_\alpha \uparrow$ |
|---|---|---|---|---|
| PVT w/o Cam | 0.022 | 0.805 | 0.938 | 0.871 |
| PVT w/o pretrain | 0.022 | 0.807 | 0.940 | 0.870 |
| PVT w/ pretrain | **0.021** | **0.811** | **0.939** | **0.873** |

(a) Effect of the pre-trained model with ResNet50.  (b) Effect of the pre-trained model with PVT.

Table S8: Explorations of the Camouflageator framework, where "Cam" is short for Camouflageator. (a) R50 means ResNet50 (He et al., 2016). The best epoch of "R50 w/o pretrain" is 67 (COD10K). (b) PVT denotes PVT V2 (Wang et al., 2022). The best epoch of "PVT w/o pretrain" is 33 (COD10K). The pre-trained models help Camouflageator to better improve segmentation performance.

| Methods | $M \downarrow$ | $F_\beta \uparrow$ | $E_\phi \uparrow$ | $S_\alpha \uparrow$ |
|---|---|---|---|---|
| w/o discriminator | **0.028** | **0.763** | **0.920** | **0.843** |
| w/ discriminator | 0.029 | 0.758 | 0.915 | 0.840 |

Table S9: Effect of the discriminator.

Additionally, we also illustrate the feasibility of corrupting the edge information in a weighted manner. Specifically, we substitute the unweighted edge mask $\mathbf{y}_e^1$ for the weighted edge mask $\mathbf{y}_e$ utilized in $L_{tc}$ and reformulate the concealment loss, termed $L_g^{c2}$:

$$
\begin{aligned}
L_g^{c2} = L_f &+ \lambda \|\mathbf{y} \otimes \mathbf{x}_g - P_o^I\|^2 \\
&+ \lambda \|\mathbf{y}_e^1 \otimes \mathbf{x}_g - P_e^{I1}\|^2,
\end{aligned}
\tag{2}
$$

where $P_e^{I1}$ is the image-level edge prototype which is an average of edge pixels specified by $\mathbf{y}_e^1$.

As shown in Table S2, the detector trained with $L_g^{c1}$ or $L_g^{c2}$ also achieves performance gains, but the improvements are not as significant as those obtained when trained with $L_g^c$. These results comprehensively verify the advancement of our fidelity loss $L_f$ and concealment loss $L_{cl}$.

**Performance at different resolutions.** To investigate the impact of the image resolution on ICEG, we evaluate the segmentation results of our ICEG with various image resolutions. As illustrated in Table S3, we observe that performance improves as the image resolution increases. This discovery motivates us to incorporate image super-resolution techniques into the COD task in future research.

**Effect of ASPP.** We employ ASPP to enlarge the receptive field, fuse the multi-context information, and generate a coarse segmentation result to guide the subsequent segmentation, whose effectiveness is demonstrated in Table S4.

**Effect of camouflaged consistency loss.** Apart from focusing on feature correlations as in existing multi-scale frameworks, we design a novel camouflaged consistency loss $L_{cc}$ to address incomplete segmentation by enhancing the internal consistency of camouflaged objects. As shown in Table S5, we incorporate $L_{cc}$ to existing multi-scale frameworks (Jia et al., 2022; Zhang et al., 2022a; Zhai et al., 2022) and find improved performance, thereby confirming the superiority of $L_{cc}$.

**Comparisons with other edge-based methods.** As mentioned in the Related work, ICEG stands apart from existing edge-based detectors (Zhai et al., 2021; Sun et al., 2022) by utilizing edge guidance that is updated with the segmentation module from coarse to fine to adaptively guide segmentation under the separated attentive framework. The superiority of our edge-based ICEG is verified in Table S6, where we replace the edge modules in MGL (Zhai et al., 2021) and BG-Net (Sun et al., 2022) with ESD and observe an evident improvement in segmentation results.

**Effect of ESC.** ESC is employed to explicitly guide the segmentation with edge information. In Table S7, we conduct an experiment to verify the superiority of ESC, where (*,*,*,*) means whether

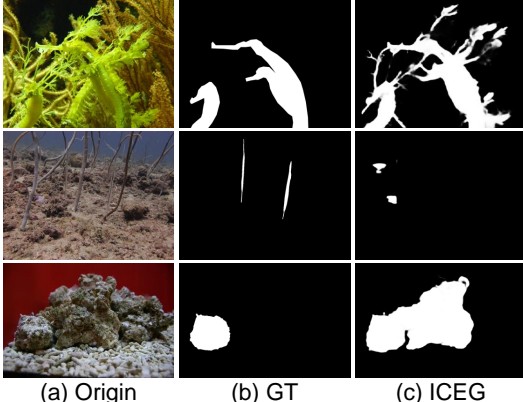

(a) Origin          (b) GT          (c) ICEG

Figure S1: Failure cases of ICEG.

ESC is retained in $D_k$, e.g., (1,0,0,0) indicates ESC is retained in $D_4$ but not in the other three decoders. As shown in Table S7, ESC can promote better segmentation results, and the current version in ICEG, namely (1,1,1,1), achieves the optimal performance.

**Discussions of Camouflageator.** In this discussion, we explore two problems related to Camouflageator, namely "*How to effectively employ the Camouflageator framework?*" and "*How to deal with the domain shift problem caused by synthesized images?*". To address the first problem, we conduct ablations for detectors (see Tables S8a and S8b), and find that using a pre-trained detector can improve the segmentation performance. In Tables S8a and S8b, we train the detector with no pre-trained model by 130 epochs and report the best results within the 130 epochs. As reported in Tables S8a and S8b, ICEG with the pretrain model achieves better results in the backbones of ResNet50 and PVT V2 and thus indicates the value of the pre-trained model. This lies in the fact that a naive detector cannot effectively learn from the synthesized challenging images and fails to give valuable feedback to the generator, thereby affecting training stability and segmentation results.

Regarding to the second problem, the purpose of the Camouflageator framework is to strengthen the generalizability of the detector, not to ensure that the distribution of the synthesized images is completely unbiased from the original camouflaged data. Focusing too much on the latter may even cause a degradation of segmentation performance, which is verified by Table S9, where we introduce a classification-based discriminator (Deng et al., 2022) for the generator to enforce the "realism" of the synthesized images. However, this degrades performance because the generator needs to balance the requirements of the discriminator while also accommodating the detector. Additionally, we have already ensured the quality of the generated camouflaged images by employing the fidelity loss $L_f$ to constrain the visually consistent between the generated image and the original image, and we only fine-tune the pre-trained detector by 30 epochs to enhance the generalizability of the detector without overfitting to the synthesized concealed images.

## C LIMITATIONS AND FUTURE WORK

**Camouflageator.** As shown in Table S8b, owing to the strong feature extraction capacity of the transformer, Camouflageator can only bring limited improvements for ICEG with the backbone of PVT. In future work, we plan to propose more powerful generators to cater to those transformer-based backbones by further reinforcing their generalizability. Additionally, we also consider extending our framework to other tasks such as saliency object detection, industrial defect detection, medical image segmentation, etc.

**ICEG.** As depicted in Fig. S1, ICEG may fail to accurately identify those concealed objects that share very similar structural information to the background. To handle this, we consider extending our method in the following aspects to cope with this problem: (1) We intend to employ a coarse-to-fine framework, wherein we can first detect those candidates and then finetune the coarse maps to ensure accurate segmentation. (2) we aim to enhance the feature extraction capacity to make sure that we can better filter out those valuable objects. Furthermore, we will consider utilizing our segmentor to extract more semantic-level information (Zhang et al., 2021; Xiao et al., 2024; Pu et al.,

2023a; Xiao et al., 2023) and improve the generalizability to empower more fields (Fang & Han, 2023; Ma et al., 2023; Tang et al., 2023). Additionally, we plan to employ self-excavation strategies to mine the valuable information (Zhang et al., 2022b; Xu et al., 2022) or incorporating more powerful architectures, e.g., dynamic networks (Pu et al., 2023c; Wang et al., 2021), transformer (Guo et al., 2022; Pu et al., 2023b; Ma et al., 2022), and diffusion model (Ma et al., 2023; Li et al., 2023; Guo et al., 2023b), with more strategic pretrain networks (Xu et al., 2023b; Ni et al., 2023; Guo et al., 2023a). (3) We consider ensuring the generalizability of our segmentor even under extremely degraded scenarios. First, we think it would be desirable to employ image quality assessment techniques (Hu et al., 2021a;b) to distinguish hard samples, which helps us to focus more on those valuable samples. Furthermore, fusing multi-modality data, *e.g.*, infrared and visible image fusion, can greatly improve the accuracy of downstream tasks (Xu et al., 2023a; Ju et al., 2022). Therefore, multi-modality image fusion can be a good solution to address camouflaged object detection. Moreover, how to train a segmentor to directly cope with those degraded scenarios, including low-light scenarios and hazy environments, is a valuable direction that is worth exploring. In this case, the future design of our segmentor should also refer to low-quality image restoration techniques (He et al., 2023a; Ye et al., 2023; 2022) to improve the global illumination and texture details.

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
