# OpenReview forum: "Strategic Preys Make Acute Predators: Enhancing Camouflaged Object Detectors by Generating Camouflaged Objects"
_ICLR.cc/2024/Conference — ICLR 2024 poster_

### Official Review · Reviewer_pcqB · 2023-10-29

**Soundness:** 3 good
**Presentation:** 3 good
**Contribution:** 3 good
**Rating:** 8
**Confidence:** 5

**Summary:**

This paper presents an adversarial training framework, Camouflageator, generating more yet more challenging camouflaged objects to enhance generalizability. Additionally, it proposes a new COD method, ICEG to tackle the incomplete segmentation and ambiguous boundary limitations of existing methods.

**Strengths:**

- The inspiration from the prey-vs-predator game provides an interesting and effective COD method.
- Designed a 2-phase training pipeline combining the generation and detection.
- Proposed a Camouflageator with flexibility and generalizability that could be applied to various existing COD detectors.
- Proposed an ICEG detector including a CFC module and an ESC module which leads to a better segmentation quality.
- Achieved the state-of-the-art on four benchmarks.

**Weaknesses:**

- The paper mentions the proposed Camouflageator generates "more camouflaged objects that are harder for COD detectors" many times. However, the paper does not include a detailed description of the quantity or quality of the synthesized camouflaged objects.
- Though the generalizability of Camouflageator has been validated with ResNet50 based COD methods, the paper does not conduct experiments with other COD methods employing Res2Net50 and Swin as backbones.
- Though Fig 5 indicates the ICEG has better segmentation quality in terms of completeness and boundaries compared to other methods, they do not provide the comparison (visualization) of how each module improves the backbone.

**Questions:**

- Please provide related statistics about the synthesized camouflaged objects? For example, the number of categories and the camouflaged property (e.g., color, contrasts, intensity comparisons of foreground and background).
- Please supplement the experiments regarding the Camouflageator generalizability with Res2Net50 and Swin based methods?
- Please provide the visualization maps of module ablations, and consider leveraging CAM-family (activation maps) for better illustration.

---

> ### Author Response · Authors · 2023-11-20
> **Rebuttal by Authors 1**
>
> Thanks for the valuable comments.
>
> **Q1: Descriptions of the quantity or quality of the synthesized camouflaged objects.**
>
> **Quantitative analysis**: There are three critical aspects to consider in maintaining the undetectability of a camouflaged object: (1) internal consistency (IC); (2) edge disruption (ED); (3) environment adaptation (EA). To assess the camouflaged level, accounting for color and contrast, we have designed three corresponding metrics:
>
> $IC = \|\mathbf{y} \otimes \mathbf{x}_g-P_o^{I}\|^2,$
>
> $ED = \| \mathbf{y}_e \otimes \mathbf{x}_g - P_e^{I} \|^2,$
>
> $EA = \|\mathbf{x}_g-P_g^{I}\|^2,$
>
> where $\mathbf{x}_g$ is the synthesized camouflaged image, $\mathbf{y}$ is the ground truth binary mask, $\otimes$ denotes element-wise multiplication, $\mathbf{y}_e$ is the weighted edge mask dilated by Gaussian function. $P_o^{I}$ is the image-level object prototype, averaging foreground pixels. $P_e^{I}$ is the image-level edge prototype, averaging edge pixels specified by $\mathbf{y}_e$. $P_g^{I}$ is the image-level object prototype, averaging global pixels.
>
> For clarity and comparability, we normalize these three metrics between [0,1] and **smaller values** indicate more covert camouflaged objects. It's essential to note that the generation of additional camouflaged objects occurs exclusively on the training set. Consequently, we evaluate the enhanced camouflaged images and the original camouflaged images solely on the training set. As shown in the table, the enhanced camouflaged images generated by our Camouflageator are more deceptive than the original images.
>
> |Datasets|CAMO|(1000 |images)|COD10K|(3040 |images)|
> |-|-|-|-|-|-|-|
> ||IC|ED|EA|IC|ED|EA|
> |Original|0.415|0.627|0.339|0.508|0.609|0.348|
> |Enhanced|0.086|0.092|0.103|0.096|0.102|0.091|
>
> **Qualitative analysis**: As shown in Fig. 6 in the manuscript, our Camouflageator can generate more concealed objects by hiding discriminative information. For example, the mouth and eyes of the fish are deliberately hidden in our synthesized image, and the physiological information of the rabbit is also weakened.
>
> **Q2: Visualization of module ablations.**
>
> We have included visualization maps of module ablations in Fig. S2 of the supplementary material. To enhance clarity, we have updated this figure using the CAM-family for better illustration. As exhibited in Fig. 2(d), the detector with CFC module can better excavate the internal coherence of camouflaged objects, facilitating the generation of more complete prediction maps. Additionally, as depicted in Fig. 2(e), the network equipped with the ESD component can decrease the uncertainty fringes and thus generate more precise results with clearer boundaries.

---

> ### Author Response · Authors · 2023-11-20
> **Rebuttal by Authors 2**
>
> **Q3: Verify the generalizability of Camouflageator on the COD methods with Res2Net50 and Swin backbones.**
>
> We further assess the generalizability of Camouflageator on detectors with various backbones, including Res2Net50-based, Swin Transformer-based, and PVT V2-based detectors. We consider two conditions: (1) the backbone of the auxiliary generator adopts ResNet50; (2) the backbone of the auxiliary generator aligns with the detector. If the original method is denoted as ICEG, we term the methods optimized in condition 1 and condition 2 as ICEG+ and ICEG++, respectively. Results presented in the table reveal that Camouflageator consistently enhances the performance of these detectors in both conditions, with greater improvement observed when the generator shares the same backbone with the detector. Specifically, Camouflageator improves ICEG by 2.9%, 3.2%, and 4.3% when the backbone of the generator and the detector are Res2Net50, Swin Transformer, and PVT V2. Furthermore, Camouflageator enhances BGNet, a state-of-the-art Res2Net50-based detector, by 2.3% (BGNet+ with ResNet50-based generator) and 2.7% (BGNet++ with Res2Net50-based generator). For the Swin Transformer-based detector, Camouflageator improves ICON by 1.5% (ICON+ with ResNet50-based generator) and 3.7% (ICON++ with Swin Transformer-based generator). Additionally, HitNet, a PVT V2-based detector, exhibits improvements with our Camouflageator, showing an increase of 0.9% (HitNet+ with ResNet50-based generator) and 3.9% (HitNet++ with PVT V2-based generator).
>
> |Methods||Backbones|*CHAMELEON*||||*CAMO*||||*COD10K*||||*NC4K*||||
> |-|-|-|-|-|-|-|-|-|-|-|-|-|-|-|-|-|-|-|
> ||||$M$|$F_\beta$|$E_\phi$|$S_\alpha$|$M$|$F_\beta$|$E_\phi$|$S_\alpha$|$M$|$F_\beta$|$E_\phi$|$S_\alpha$|$M$|$F_\beta$|$E_\phi$|$S_\alpha$|
> |BGNet||Res2Net50|0.029|0.835|0.944|0.895|0.073|0.744|0.870|0.812|0.033|0.714|0.901|0.831|0.044|0.786|0.907|0.851|
> |BGNet+||Res2Net50|0.027|0.845|0.952|0.898|0.070|0.760|0.882|0.818|0.031|0.729|0.913|0.838|0.042|0.800|0.916|0.860|
> |BGNet++ (Res2Net50-based generator)||Res2Net50|0.027|0.847|0.956|0.903|0.070|0.761|0.885|0.820|0.030|0.735|0.918|0.833|0.042|0.805|0.922|0.863|
> |ICEG||Res2Net50|0.025|0.869|0.958|0.908|0.066|0.808|0.903|0.838|0.028|0.752|0.914|0.845|0.042|0.828|0.917|0.867|
> |ICEG+||Res2Net50|0.023|0.873|0.960|0.910|0.064|0.826|0.912|0.845|0.026|0.770|0.925|0.853|0.040|0.844|0.928|0.878|
> |ICEG++ (Res2Net50-based generator)||Res2Net50|0.023|0.876|0.966|0.915|0.063|0.833|0.916|0.849|0.026|0.777|0.930|0.855|0.039|0.847|0.932|0.879|
> |ICON||Swin|0.029|0.848|0.940|0.898|0.058|0.794|0.907|0.840|0.033|0.720|0.888|0.818|0.041|0.817|0.916|0.858|
> |ICON+||Swin|0.028|0.855|0.943|0.899|0.056|0.808|0.915|0.843|0.031|0.728|0.895|0.822|0.040|0.823|0.922|0.861|
> |ICON++ (Swin transformer-based generator)||Swin|0.026|0.862|0.949|0.902|0.055|0.823|0.926|0.849|0.029|0.749|0.906|0.830|0.037|0.836|0.931|0.866|
> |ICEG||Swin|0.023|0.860|0.959|0.905|0.044|0.855|0.926|0.867|0.024|0.782|0.930|0.857|0.034|0.855|0.932|0.879|
> |ICEG+||Swin|0.022|0.867|0.961|0.908|0.042|0.861|0.931|0.871|0.023|0.788|0.934|0.862|0.033|0.861|0.937|0.883|
> |ICEG++ (Swin transformer-based generator)||Swin|0.021|0.883|0.969|0.911|0.041|0.868|0.938|0.876|0.021|0.803|0.945|0.865|0.032|0.869|0.944|0.885|
> |HitNet||PVT|0.024|0.861|0.944|0.907|0.060|0.791|0.892|0.834|0.027|0.790|0.922|0.847|0.042|0.825|0.911|0.858|
> |HitNet+||PVT|0.023|0.863|0.945|0.909|0.060|0.797|0.895|0.835|0.026|0.794|0.925|0.849|0.041|0.829|0.916|0.860|
> |HitNet++ (PVT-based generator)||PVT|0.022|0.879|0.957|0.916|0.057|0.820|0.913|0.839|0.023|0.818|0.936|0.853|0.037|0.849|0.931|0.867|
> |ICEG||PVT|0.022|0.879|0.957|0.913|0.043|0.863|0.933|0.876|0.022|0.805|0.938|0.871|0.030|0.869|0.941|0.890|
> |ICEG+||PVT|0.022|0.881|0.958|0.915|0.041|0.867|0.935|0.877|0.021|0.811|0.939|0.873|0.029|0.875|0.944|0.893|
> |ICEG++ (PVT-based generator)||PVT|0.021|0.889|0.965|0.920|0.039|0.880|0.949|0.881|0.019|0.832|0.968|0.903|0.025|0.896|0.960|0.908|

---

> ### Author Response · Authors · 2023-11-23
> **Official Comment by Authors**
>
> Dear Reviewer pcqB,
>
> We appreciate the reviewer’s time and effort in reviewing our manuscript and insightful comments.
>
> As the closure of the discussion period is approaching, we would like to bring the review’s attention and check if the reviewer could let us know whether the concerns or the misunderstanding have been addressed.
>
> If this is the case, we would appreciate if you could adjust your rating accordingly.
>
> Best regards,
>
> Authors

---

> > ### Comment · Reviewer_pcqB · 2023-11-23
> >
> > I appreciate the authors' response and effort.
> >
> > The intensive experiments conducted as well as relevant analysis do address my concerns.
> >
> > I am willing to maintain my previous rating.

---

> > > ### Author Response · Authors · 2023-11-23
> > > **Thanks for recognizing the value of our work!**
> > >
> > > We extend our gratitude to the reviewer for acknowledging the significance of our contribution, namely the plug-and-play **Camouflageator** framework and the proficiently executed **ICEG** detector, within the domain of camouflaged object detection (COD). Your recognition is deeply appreciated and serves as validation of our efforts in advancing this critical area of research.

---

### Official Review · Reviewer_bgwW · 2023-11-02

**Soundness:** 3 good
**Presentation:** 3 good
**Contribution:** 2 fair
**Rating:** 5
**Confidence:** 1

**Summary:**

The paper proposed to address COD on both the prey and predator sides. On the prey side, it introduced a novel adversarial training strategy, Camouflageator, to enhance the generalizability of the detector by generating more camouflaged objects harder for a COD detector to detect. On the predator side, it designed a novel detector, dubbed ICEG, to address the issues of incomplete segmentation and ambiguous boundaries.

**Strengths:**

++ An adversarial training framework, Camouflageator, for the COD task to employ an auxiliary generator that generates more camouflaged objects that are harder for COD detectors to detect and hence enhances the generalizability of those detectors. Camouflageator is flexible and can be integrated with various existing COD detectors.

++ A new COD detector, ICEG, to address the issues of incomplete segmentation and ambiguous boundaries that existing detectors face. ICEG introduces a novel CFC module to excavate the internal coherence of camouflaged objects to obtain complete segmen-tation results, and an ESC module to leverage edge information to get precise boundaries.

**Weaknesses:**

-- “CamDiff: Camouflage Image Augmentation via Diffusion Model” also utilizes the generation of camouflaged images to help with COD and should be included to the comparison.

-- There exist many COD methods are not mentioned in  RELATED WORK: CAMOUFLAGED OBJECT DETECTION part, like:
Mutual graph learning for camouflaged object detection
A Bayesian approach to camouflaged moving object detection
Uncertainty-guided transformer reasoning for camouflaged object detection
Deep texture-aware features for camouflaged object detection
Although some papers are cited, they need to be reflected in the related work

-- Why does Equation 4 use two losses for segmentation.

-- In Sec 3.2.1 Camouflaged consistency loss, what is the difference between the ideas in the paper and the contrastive loss

-- In Eq 17, are mu and sigma the same?
-- Why backbone chose resnet50 and not the transformer structure

**Questions:**

Please address my major concerns as listed in the Weaknesses section.

---

> ### Author Response · Authors · 2023-11-20
> **Rebuttal by Authors 1**
>
> Thanks for the valuable comments. If not specifically stated, all experiments are conducted on the COD10K dataset for space limitation.
>
> **Q1: Difference with CamDiff.**
>
> Idea: The motivation behind CamDiff is to address the challenge of existing COD methods in distinguishing between salient and camouflaged objects. Conversely, our approach focuses on enhancing the discriminative feature extraction capacity of current detectors to improve their overall generalizability. Since the root issue CamDiff aims to resolve is the insufficient discriminative information extraction by existing detectors between salient and camouflaged objects, our strategy can also effectively address this problem.
>
> Implementation: CamDiff generates synthetic salient objects within camouflaged scenes and establishes a **fixed** dataset for detector training. In contrast, our Camouflageator employs a continuously updated generator, synthesizing training data while simultaneously training the detector in an adversarial manner. This approach **tailors** the dataset for each detector, enhancing flexibility and ensuring superior performance.
>
> Experiment: The results presented in the table compare detectors trained with our Camouflageator and CamDiff. Notably, detectors optimized by Camouflageator consistently achieve superior performance. In contrast, detectors optimized by CamDiff occasionally exhibit lower performance than their original versions. This discrepancy may stem from the introduction of salient objects disrupting overall image harmonization during CamDiff's dataset construction. Moreover, the second row of Fig. S2 in the supplementary material illustrates that the original detector, ICEG, encounters challenges in separating salient and camouflaged objects. However, after being optimized by the Camouflageator, ICEG+ adeptly identifies camouflaged objects. This underscores that Camouflageator effectively addresses the problem CamDiff seeks to solve, yielding substantial performance improvements.
>
> |Methods|COD10K||||
> |-|-|-|-|-|
> ||$M$|$F_\beta$|$E_\phi$|$S_\alpha$|
> |FGANet|0.032|0.708|0.894|0.803|
> |FGANet+CamDiff|0.032|0.711|0.886|0.807|
> |FGANet+ (Ours)|0.030|0.735|0.911|0.823|
> |FEDER|0.032|0.715|0.892|0.810|
> |FEDER+CamDiff|0.033|0.704|0.881|0.805|
> |FEDER+ (Ours)|0.030|0.739|0.905|0.831|
> |ICEG|0.030|0.747|0.906|0.826|
> |ICEG+CamDiff|0.031|0.722|0.898|0.821|
> |ICEG+ (Ours)|0.028|0.763|0.920|0.843|
>
>  **Q2: More related works.**
>
> We add the discussions of those papers in the Related Works. Notice that our related work provides a comprehensive categorization of the types of learning-based methods available and that the three learning-based methods can also be classified into these three categories. For instance, MGL [1] falls into the category utilizing a joint training strategy. Similarly, UGTR [2] and DTAF [3] can be classified under the category employing multi-scale feature aggregation.
>
> **Q3: Losses used for segmentation.**
>
> It is a common practice to use weighted BCE loss and weighted IoU loss for supervision in the field of COD, and our approach aligns with this convention. Weighted BCE loss treats each pixel as an independent sample for prediction, whereas weighted IoU loss provides a more comprehensive evaluation of the final prediction. The synergy between these two loss functions ensures a thorough assessment of the prediction results.
>
>  **Q4: Difference between camouflaged consistency loss and contrastive loss.**
>
> The contrastive loss emphasizes the alignment of positive pairs (and misalignment of negative pairs). As a difference, our camouflaged consistency loss simultaneously encourages **intra-group compactness** and **inter-group separation**. Our loss is particularly designed for camouflaged image pixels. In this context, foreground pixels exhibit a strong intrinsic correlation, while the composition of background pixels tends to be intricate. As a result, our camouflaged consistency loss exclusively constrains the foreground pixels. We aim for the foreground pixels to exhibit compactness among themselves, and we further desire these compact foreground pixels to be distinct from the background pixels. Notably, we intentionally avoid enforcing compactness on background pixels due to their inherent redundancy, as such a requirement could compromise the feature extraction capability.
>
> **Q5: Are $\mu$ and $\sigma$ the same?**
>
> $\mu$ and $\sigma$ are variational parameters and they are different. We reformulate Eq. 17 to avoid misunderstanding.
>
> [1] MGL, CVPR21.
>
> [2] UGTR, ICCV21.
>
> [3] DTAF, TCSVT.

---

> ### Author Response · Authors · 2023-11-20
> **Rebuttal by Authors 2**
>
> **Q6: Why resnet50 rather than transformer?**
>
> We choose ResNet50 as the backbone of the auxiliary generator, aligning with common practice in generator design. It's important to note that our Camouflageator framework is not confined to specific models. We conducted a thorough evaluation of Camouflageator's generalizability across generators with diverse backbones, including Res2Net50-based, Swin Transformer-based, and PVT V2-based generators. In exploring the suitability of different backbones for Camouflageator, our experiments focused on two conditions: (1) the auxiliary generator's backbone is ResNet50, and (2) the auxiliary generator shares the same backbone with the detector. Denoting the original method as ICEG, we term the methods optimized in conditions 1 and 2 as ICEG+ and ICEG++, respectively. The results presented in the table consistently demonstrate that Camouflageator enhances the performance of these detectors under both conditions, with more significant improvement observed when the generator shares the same backbone with the detector. Specifically, Camouflageator improves ICEG by 2.9%, 3.2%, and 4.3% when the backbones of the generator and the detector are Res2Net50, Swin Transformer, and PVT V2, respectively. Furthermore, Camouflageator enhances BGNet, a state-of-the-art Res2Net50-based detector, by 2.3% (BGNet+ with ResNet50-based generator) and 2.7% (BGNet++ with Res2Net50-based generator). For the Swin Transformer-based detector, Camouflageator improves ICON by 1.5% (ICON+ with ResNet50-based generator) and 3.7% (ICON++ with Swin Transformer-based generator). Additionally, HitNet, a PVT V2-based detector, shows improvements with our Camouflageator, exhibiting an increase of 0.9% (HitNet+ with ResNet50-based generator) and 3.9% (HitNet++ with PVT V2-based generator).
>
> |Methods||Backbones|*CHAMELEON*||||*CAMO*||||*COD10K*||||*NC4K*||||
> |-|-|-|-|-|-|-|-|-|-|-|-|-|-|-|-|-|-|-|
> ||||$M$|$F_\beta$|$E_\phi$|$S_\alpha$|$M$|$F_\beta$|$E_\phi$|$S_\alpha$|$M$|$F_\beta$|$E_\phi$|$S_\alpha$|$M$|$F_\beta$|$E_\phi$|$S_\alpha$|
> |BGNet||Res2Net50|0.029|0.835|0.944|0.895|0.073|0.744|0.870|0.812|0.033|0.714|0.901|0.831|0.044|0.786|0.907|0.851|
> |BGNet+||Res2Net50|0.027|0.845|0.952|0.898|0.070|0.760|0.882|0.818|0.031|0.729|0.913|0.838|0.042|0.800|0.916|0.860|
> |BGNet++ (Res2Net50-based generator)||Res2Net50|0.027|0.847|0.956|0.903|0.070|0.761|0.885|0.820|0.030|0.735|0.918|0.833|0.042|0.805|0.922|0.863|
> |ICEG||Res2Net50|0.025|0.869|0.958|0.908|0.066|0.808|0.903|0.838|0.028|0.752|0.914|0.845|0.042|0.828|0.917|0.867|
> |ICEG+||Res2Net50|0.023|0.873|0.960|0.910|0.064|0.826|0.912|0.845|0.026|0.770|0.925|0.853|0.040|0.844|0.928|0.878|
> |ICEG++ (Res2Net50-based generator)||Res2Net50|0.023|0.876|0.966|0.915|0.063|0.833|0.916|0.849|0.026|0.777|0.930|0.855|0.039|0.847|0.932|0.879|
> |ICON||Swin|0.029|0.848|0.940|0.898|0.058|0.794|0.907|0.840|0.033|0.720|0.888|0.818|0.041|0.817|0.916|0.858|
> |ICON+||Swin|0.028|0.855|0.943|0.899|0.056|0.808|0.915|0.843|0.031|0.728|0.895|0.822|0.040|0.823|0.922|0.861|
> |ICON++ (Swin transformer-based generator)||Swin|0.026|0.862|0.949|0.902|0.055|0.823|0.926|0.849|0.029|0.749|0.906|0.830|0.037|0.836|0.931|0.866|
> |ICEG||Swin|0.023|0.860|0.959|0.905|0.044|0.855|0.926|0.867|0.024|0.782|0.930|0.857|0.034|0.855|0.932|0.879|
> |ICEG+||Swin|0.022|0.867|0.961|0.908|0.042|0.861|0.931|0.871|0.023|0.788|0.934|0.862|0.033|0.861|0.937|0.883|
> |ICEG++ (Swin transformer-based generator)||Swin|0.021|0.883|0.969|0.911|0.041|0.868|0.938|0.876|0.021|0.803|0.945|0.865|0.032|0.869|0.944|0.885|
> |HitNet||PVT|0.024|0.861|0.944|0.907|0.060|0.791|0.892|0.834|0.027|0.790|0.922|0.847|0.042|0.825|0.911|0.858|
> |HitNet+||PVT|0.023|0.863|0.945|0.909|0.060|0.797|0.895|0.835|0.026|0.794|0.925|0.849|0.041|0.829|0.916|0.860|
> |HitNet++ (PVT-based generator)||PVT|0.022|0.879|0.957|0.916|0.057|0.820|0.913|0.839|0.023|0.818|0.936|0.853|0.037|0.849|0.931|0.867|
> |ICEG||PVT|0.022|0.879|0.957|0.913|0.043|0.863|0.933|0.876|0.022|0.805|0.938|0.871|0.030|0.869|0.941|0.890|
> |ICEG+||PVT|0.022|0.881|0.958|0.915|0.041|0.867|0.935|0.877|0.021|0.811|0.939|0.873|0.029|0.875|0.944|0.893|
> |ICEG++ (PVT-based generator)||PVT|0.021|0.889|0.965|0.920|0.039|0.880|0.949|0.881|0.019|0.832|0.968|0.903|0.025|0.896|0.960|0.908|

---

> ### Author Response · Authors · 2023-11-23
> **Official Comment by Authors**
>
> Dear Reviewer bgwW,
>
> We appreciate the reviewer’s time and effort in reviewing our manuscript and insightful comments.
>
> As the closure of the discussion period is approaching, we would like to bring the review’s attention and check if the reviewer could let us know whether the concerns or the misunderstanding have been addressed.
>
> If this is the case, we would appreciate if you could adjust your rating accordingly.
>
> Best regards,
>
> Authors

---

### Official Review · Reviewer_VfAG · 2023-11-05

**Soundness:** 2 fair
**Presentation:** 3 good
**Contribution:** 2 fair
**Rating:** 5
**Confidence:** 2

**Summary:**

The present paper proposes an innovative framework for the COD task based on adversarial training. While adversarial training has been established as an effective method in domains such as image classification, image/video deblur and object detection, this study successfully implemented it for the COD task. The efficacy of the proposed method is demonstrated through extensive experimentation on COD benchmarks.

**Strengths:**

1. The paper generally well written.
2. The idea of using an adversarial generator to create more difficult training examples addresses an important weakness (lack of diversity) in existing COD datasets.
3. ICEG's modules for improving segmentation completeness and boundary precision tackle the limitations of prior work. The design choices are well-motivated.
4. Extensive experiments demonstrate SOTA results on multiple datasets. Ablations verify the contributions of individual components.

**Weaknesses:**

1. The paper presents two significant contributions: firstly, the introduction of an adversarial training framework for COD tasks, and secondly, the implementation of ICEG to overcome the limitations of incomplete segmentation and ambiguous boundaries for camouflaged objects. However, the two contributions seem disconnected, and they appear to be independent approaches aimed at enhancing network segmentation without a clear connection between them. Consequently, the primary storyline of the work is missing. The authors must establish the relationship between the two contributions and articulate how they work together to achieve the overall objective of the paper.

2. The adversarial training framework, which represents the primary contribution of this paper, is not a novel concept. Although it has yet to be implemented in COD tasks, the fundamental idea and implementation are similar to other tasks, such as image deblurring. It would be beneficial to underscore the unique difficulties and differences between COD and other tasks if the implementation is non-trivial.

3. Missing detailed descriptions for Figure 3. It lacks detailed descriptions and makes it challenging to comprehend the main idea presented. Given the complexity of the figure, it is important to provide proper explanations to ensure a comprehensive understanding. I think that the explanation of the network design may come across as a technical report and, as such, should be included in the supplementary material. In the body of the paper, I would like to see clear and concise motivations and rationales for each design, rather than technical details. The author should enhance the descriptions in Figure 3 to ensure they are easily understandable for readers.

4. It is imperative to ascertain the efficacy of the proposed adversarial training framework for other state-of-the-art (SOTA) networks by conducting ablations of the framework with alternative methods. This endeavor would furnish evidence to substantiate the generalizability of the proposed approach and its applicability to other networks.

**Questions:**

N/A

**Details Of Ethics Concerns:**

The proposed methodology appears to offer potential utility in military contexts and, as such, necessitates a rigorous ethical review. It is imperative that all ethical considerations are carefully considered and addressed, in order to ensure that the proposed method is implemented in a manner that aligns with established ethical principles. Therefore, a comprehensive analysis of the proposed methodology, as well as its potential implications, is essential for the development of a comprehensive and responsible implementation plan.

---

> ### Author Response · Authors · 2023-11-20
> **Rebuttal by Authors 1**
>
> Thanks for the valuable comments.
>
> **Q1: Missing primary storyline and lacking the establishment of the relationship between Camouflageator and ICEG.**
>
> As stated in the third paragraph of the Introduction, our method is inspired by the prey-vs-predator game, which leads to ever-strategic preys and ever-acute predators, and we propose to address COD by developing algorithms on both the prey side (Camouflageator) that generates more deceptive camouflage objects and the predator side (ICEG) that produces complete and precise detection results. Therefore, Camouflageator and ICEG are complementary and can be combined.
>
> In Section 3.2.4, we present how to integrate the Camouflageator framework with ICEG and denote the integrated method as ICEG+. As depicted in Fig. 1 in the manuscript, ICEG+ successfully achieves the overarching objectives of our paper and excels in addressing the extreme challenges in COD, including false localization, incomplete prediction, and ambiguous boundaries.
>
> In the updated manuscript, we have revised the Introduction to make it clearer about the relationship between Camouflageator and ICEG. Thank you for pointing it out.
>
> **Q2: Difference of Camouflageator between the adversarial training framework in other tasks.**
>
> Our method diverges significantly from practices employed in other tasks, such as image deblurring, both in fundamental idea and implementation.
>
> Fundamental idea: Existing methods, such as image deblurring methods [1,2,3], typically utilize adversarial training strategies, employing a classification-based discriminator to **ensure the generation of visually faithful results**. The focus lies on the **generator** side.  In contrast, drawing inspiration from the prey-vs-predator game, we introduce a task-oriented framework, Camouflageator, for COD. The primary objective is to generate more deceptive camouflage objects, thereby **enhancing the generalizability of the detector** (termed task-oriented). Our focus lies on the **detector** (or discriminator) side.
>
> Implementation: To achieve the goal of deception, we introduce the concealment loss (Eq. 3) for generating deceptive objects, a concept not previously proposed in existing adversarial training methods. Unlike purely data-driven generators (task-agnostic), our auxiliary generator is partially supervised by the detector (Eq. 4) in a task-wise manner. Additionally, in contrast to prevailing adversarial training strategies, we deliberately modify the network and omit the discriminator (refer to Table S11 in the supplementary material). This choice which is deliberate as the objective of our Camouflageator framework is to bolster the generalizability of the detector, not to ensure complete unbiasedness of the synthesized images' distribution from the original camouflaged data. Overemphasis on the latter aspect may even lead to a degradation of detection performance.
>
> **Q3: Description of Fig. 3.**
>
> The input and output of Fig.3 are a camouflaged image and the corresponding prediction result generated by our ICEG. The proposed ICEG mainly has two critical components: the camouflaged feature coherence (CFC) and the edge-guided segmentation decoder (ESD). CFC employs the intra-layer feature aggregation (IFA), the contextual feature aggregation (CFA), and a camouflaged consistency loss $L_{cc}$ to alleviate incomplete segmentation. ESD comprises an edge reconstruction (ER) module and an edge-guided separated calibration (ESC) module to cooperatively eliminate ambiguous boundaries. We have revised the caption of Fig. 3 to make it easier to understand.
>
> We have reorganized the Methodology to present clear and concise motivations and rationales for each design.
>
> [1] Deblurgan, CVPR18.
>
> [2] RWB, CVPR20.
>
> [3] SAMD, TIP.

---

> ### Author Response · Authors · 2023-11-20
> **Rebuttal by Authors 2**
>
> **Q4: Verify the generalization of the proposed Camouflageator.**
>
> We have proved the generalizability of our Camouflageator, whose generator employs ResNet50 as the backbone, on several cutting-edge ResNet50-based detectors, including PreyNet, FGANet, and FEDER, in Table 1 in the manuscript. Camouflageator consistently enhances other detectors by 2.8% (PreyNet), 2.2% (FGANet), 2.3% (FEDER). Besides, Table S4 in the supplementary material illustrates that the network trained with Camouflageator excels in handling extreme conditions, such as small objects and multiple objects.
>
> We further assess the generalizability of Camouflageator on detectors with various backbones, including Res2Net50-based, Swin Transformer-based, and PVT V2-based detectors. We consider two conditions: (1) the backbone of the auxiliary generator adopts ResNet50; (2) the backbone of the auxiliary generator aligns with the detector. If the original method is denoted as ICEG, we term the methods optimized in condition 1 and condition 2 as ICEG+ and ICEG++, respectively. Results presented in the table reveal that Camouflageator consistently enhances the performance of these detectors in both conditions, with greater improvement observed when the generator shares the same backbone with the detector. Specifically, Camouflageator improves ICEG by 2.9%, 3.2%, and 4.3% when the backbone of the generator and the detector are Res2Net50, Swin Transformer, and PVT V2. Furthermore, Camouflageator enhances BGNet, a state-of-the-art Res2Net50-based detector, by 2.3% (BGNet+ with ResNet50-based generator) and 2.7% (BGNet++ with Res2Net50-based generator). For the Swin Transformer-based detector, Camouflageator improves ICON by 1.5% (ICON+ with ResNet50-based generator) and 3.7% (ICON++ with Swin Transformer-based generator). Additionally, HitNet, a PVT V2-based detector, exhibits improvements with our Camouflageator, showing an increase of 0.9% (HitNet+ with ResNet50-based generator) and 3.9% (HitNet++ with PVT V2-based generator).
>
> |Methods||Backbones|*CHAMELEON*||||*CAMO*||||*COD10K*||||*NC4K*||||
> |-|-|-|-|-|-|-|-|-|-|-|-|-|-|-|-|-|-|-|
> ||||$M$|$F_\beta$|$E_\phi$|$S_\alpha$|$M$|$F_\beta$|$E_\phi$|$S_\alpha$|$M$|$F_\beta$|$E_\phi$|$S_\alpha$|$M$|$F_\beta$|$E_\phi$|$S_\alpha$|
> |BGNet||Res2Net50|0.029|0.835|0.944|0.895|0.073|0.744|0.870|0.812|0.033|0.714|0.901|0.831|0.044|0.786|0.907|0.851|
> |BGNet+||Res2Net50|0.027|0.845|0.952|0.898|0.070|0.760|0.882|0.818|0.031|0.729|0.913|0.838|0.042|0.800|0.916|0.860|
> |BGNet++ (Res2Net50-based generator)||Res2Net50|0.027|0.847|0.956|0.903|0.070|0.761|0.885|0.820|0.030|0.735|0.918|0.833|0.042|0.805|0.922|0.863|
> |ICEG||Res2Net50|0.025|0.869|0.958|0.908|0.066|0.808|0.903|0.838|0.028|0.752|0.914|0.845|0.042|0.828|0.917|0.867|
> |ICEG+||Res2Net50|0.023|0.873|0.960|0.910|0.064|0.826|0.912|0.845|0.026|0.770|0.925|0.853|0.040|0.844|0.928|0.878|
> |ICEG++ (Res2Net50-based generator)||Res2Net50|0.023|0.876|0.966|0.915|0.063|0.833|0.916|0.849|0.026|0.777|0.930|0.855|0.039|0.847|0.932|0.879|
> |ICON||Swin|0.029|0.848|0.940|0.898|0.058|0.794|0.907|0.840|0.033|0.720|0.888|0.818|0.041|0.817|0.916|0.858|
> |ICON+||Swin|0.028|0.855|0.943|0.899|0.056|0.808|0.915|0.843|0.031|0.728|0.895|0.822|0.040|0.823|0.922|0.861|
> |ICON++ (Swin transformer-based generator)||Swin|0.026|0.862|0.949|0.902|0.055|0.823|0.926|0.849|0.029|0.749|0.906|0.830|0.037|0.836|0.931|0.866|
> |ICEG||Swin|0.023|0.860|0.959|0.905|0.044|0.855|0.926|0.867|0.024|0.782|0.930|0.857|0.034|0.855|0.932|0.879|
> |ICEG+||Swin|0.022|0.867|0.961|0.908|0.042|0.861|0.931|0.871|0.023|0.788|0.934|0.862|0.033|0.861|0.937|0.883|
> |ICEG++ (Swin transformer-based generator)||Swin|0.021|0.883|0.969|0.911|0.041|0.868|0.938|0.876|0.021|0.803|0.945|0.865|0.032|0.869|0.944|0.885|
> |HitNet||PVT|0.024|0.861|0.944|0.907|0.060|0.791|0.892|0.834|0.027|0.790|0.922|0.847|0.042|0.825|0.911|0.858|
> |HitNet+||PVT|0.023|0.863|0.945|0.909|0.060|0.797|0.895|0.835|0.026|0.794|0.925|0.849|0.041|0.829|0.916|0.860|
> |HitNet++ (PVT-based generator)||PVT|0.022|0.879|0.957|0.916|0.057|0.820|0.913|0.839|0.023|0.818|0.936|0.853|0.037|0.849|0.931|0.867|
> |ICEG||PVT|0.022|0.879|0.957|0.913|0.043|0.863|0.933|0.876|0.022|0.805|0.938|0.871|0.030|0.869|0.941|0.890|
> |ICEG+||PVT|0.022|0.881|0.958|0.915|0.041|0.867|0.935|0.877|0.021|0.811|0.939|0.873|0.029|0.875|0.944|0.893|
> |ICEG++ (PVT-based generator)||PVT|0.021|0.889|0.965|0.920|0.039|0.880|0.949|0.881|0.019|0.832|0.968|0.903|0.025|0.896|0.960|0.908|

---

> ### Author Response · Authors · 2023-11-20
> **Rebuttal by Authors 3**
>
> **Q5: Ethics concerns.**
>
> Analysis and applications of our method: Inspired by the prey-vs-predator game, our method starts from the mimicry of animals and aims to exploit the potential value of zoology and thus contribute to species discovery. Our method aims to solve the segmentation task, termed COD, that aims to detect and segment objects seamlessly integrated into surrounding environments. Therefore, our method has various latent real-life applications [4], ranging from concealed defect detection in industry, pest monitoring] in agriculture, and polyp image segmentation in medical diagnosis, to recreational art and photo-realistic blending in art. However, our approach is not directly oriented to military scenarios because we are neither considering the complex scenarios of war nor do we involve special processing of human data, such as facial information. **Note that our method can be applied to a large variety of different use cases which are beyond the scope of this work. A careful study of the data, model, intended applications, safety, risk, bias, and societal impact is necessary before any real-world application.**
>
> Potential implications of our method: We appreciate the reviewer’s advice and add an ethical statement to the manuscript.
>
> [4] COD, CVPR20.

---

> ### Author Response · Authors · 2023-11-23
> **Official Comment by Authors**
>
> Dear Reviewer VfAG,
>
> We appreciate the reviewer’s time and effort in reviewing our manuscript and insightful comments.
>
> As the closure of the discussion period is approaching, we would like to bring the review’s attention and check if the reviewer could let us know whether the concerns or the misunderstanding have been addressed.
>
> If this is the case, we would appreciate if you could adjust your rating accordingly.
>
> Best regards,
>
> Authors

---

### Official Review · Reviewer_fgyc · 2023-11-05

**Soundness:** 3 good
**Presentation:** 3 good
**Contribution:** 3 good
**Rating:** 8
**Confidence:** 4

**Summary:**

This paper is inspired by the prey-vs-predator game, proposing novel algorithms from both the prey side and the predator side. An adversarial framework is proposed to generate more challenging camouflaged objects. While the ICEG aims to detect these objects. Experiments show that this paper achieves the best results.

**Strengths:**

1. The motivation is great. This paper takes advantage of adversarial training to generate harder camouflaged objects. This idea is novel.
2. Their proposed Camouflageator is a plug-and-play framework. It is effective for ICEG and three existing methods.
3. The experiments are very detailed and thorough.

**Weaknesses:**

To some extent, the ICEG solved the incomplete segmentation. However, it fails to identify the foreground object when concealed objects have very similar structural information to the background.

Many false positive detection results are outputted.

**Questions:**

No further question.

---

> ### Author Response · Authors · 2023-11-20
> **Rebuttal by Authors**
>
> Thanks for the valuable comments.
>
> **Q1: ICEG fails to identify the foreground object when concealed objects have very similar structural information to the background.**
>
> Accurately separating foreground objects from backgrounds with very similar structure information is a common challenge for almost all COD solvers [1,2,3]. To overcome this problem, we propose the adversarial Camouflageator framework along with the CFC and ESC modules proposed in ICEG, achieving results that are both qualitatively (Fig. 5 in the manuscript and Fig. S1 in the supplementary material) and quantitatively (Table 1 in the manuscript and Table S1 in the supplementary material) better than existing methods.
>
> While being more favorable than existing COD solvers for this challenging case, we observe that our method may encounter challenges in accurately identifying the foreground, particularly in extreme scenarios, as depicted in Fig. S3 in the supplementary material. We attribute this to the intrinsic structural similarity between the foreground and the background. In response, as a future work, we are considering extending our method in the following ways to address this issue:
> - We intend to employ a coarse-to-fine framework, wherein we can first detect those candidates and then finetune the coarse maps to ensure accurate segmentation.
> - We propose to design a multi-scale feature grouping module that first groups features at different granularities and then aggregates these grouping results. Grouping similar features encourages detection coherence, helping obtain accurate results for both single and multiple-object images.
>
> We have incorporated the above content into the "Limitations" section of the supplementary material.
>
> **Q2: Many false positive detection results are outputted.**
>
> As demonstrated in the visual verification presented in Fig. S2 in the supplementary material, the incorporation of our proposed ESD module effectively reduces both false positive and false negative regions. Furthermore, our method exhibits superior performance in both qualitative and quantitative analyses. The results in Fig. 5 in the manuscript and Fig. S1 in the supplementary material underscore our method's effectiveness in addressing the false positive problem compared to existing methods.
>
> Nonetheless, when confronted with the extreme structural similarity challenge mentioned in **Q1**, our method may still encounter some false positive regions. To address this challenge in the future, we will explore and implement the strategies discussed in **Q1**.
>
> [1] PreyNet, ACM MM22.
>
> [2] FGANet, NeurIPS23.
>
> [3] FEDER, CVPR23.

---

> ### Author Response · Authors · 2023-11-23
> **Official Comment by Authors**
>
> Dear Reviewer fgyc,
>
> We appreciate the reviewer’s time and effort in reviewing our manuscript and insightful comments.
>
> As the closure of the discussion period is approaching, we would like to bring the review’s attention and check if the reviewer could let us know whether the concerns or the misunderstanding have been addressed.
>
> If this is the case, we would appreciate if you could adjust your rating accordingly.
>
> Best regards,
>
> Authors

---

### Author Response · Authors · 2023-11-20
**Author rebuttal by authors**

We extend our sincere gratitude to all the reviewers (**R1**-fgyc, **R2**-VfAG, **R3**-bgwW, and **R4**-pcqB) for their insightful and considerate reviews, which helped us to emphasize the contributions of our approach. We are encouraged to hear that the reviewers found the work is well-motivated with good presentation (**R1**, **R2**, **R3**, **R4**), as well as the comprehensive experimental evaluation and commendable performance (**R1**, **R2**, **R4**).

We are delighted to see reviewers confirm our contributions to the field of camouflaged object detection. These encompass our novel plug-and-play adversarial training framework, Camouflageator, and the ingenious camouflaged detector, ICEG, whose modules can also enhance the performance of existing detectors.

In direct response to your thoughtful comments, we have methodically addressed each point in our individual responses, and we provide a summary here:

- We reorganized the paper and revised certain formulations to enhance clarity and comprehension.

- We added experiments to verify the generalizability and advancement of our Camouflageator.

- We designed camouflage metrics for the synthesized camouflaged images to evaluate the camouflaged level.

- We conducted an ethics statement to address the ethics concerns.

Thanks again for all of your valuable suggestions. We have updated the paper accordingly and will release our code for community study. We appreciate the reviewers' time to check our response and **hope to further discuss with the reviewers whether the concerns have been addressed or not**. If the reviewers still have any unclear parts about our work, please let us know.

---

### Meta-Review · Area_Chair_XfaQ · 2023-12-05

**Metareview:**

After discussion, this submission received two positive scores and two negative scores. After reading the paper, the review comments and the rebuttal, the AC think the major concerns are about the presentation and formulation of the two proposed modules, which is encouraged to added to the camera-ready version. The novelty of the adversarial training strategy is conformed by two reviewers.

**Justification For Why Not Higher Score:**

After discussion, concerns about the ethics. increase of false defections, and connections between two proposed modules, remain. Revisions related to these major concerns are encouraged to included into the camera-ready version.

**Justification For Why Not Lower Score:**

N/A

---

### Decision · Program_Chairs · 2024-01-16

Accept (poster)